# LLM Probability Concentration: How Alignment Shrinks the Generative Horizon

## Abstract

Despite their impressive capabilities, aligned large language models (LLMs) often generate outputs that lack diversity. What drives this consistency in the generation? We investigate this phenomenon through the lens of probability concentration in the model's output distribution. To quantify this concentration, we adopt the lens of the *Branching Factor* (BF)–the exponentiated length-averaged entropy of the model's output distribution, interpreted as the effective number of plausible next steps during generation. Our empirical analysis reveals two key findings: (1) BF often decreases as generation progresses, suggesting that LLMs become more predictable as they generate. (2) alignment tuning substantially sharpens the model's output distribution from the outset, reducing BF by a factor of 2–5 overall, and up to an order of magnitude (e.g., from 12 to 1.2) at the beginning positions. This stark reduction helps explain why aligned models often appear less sensitive to decoding strategies. Building on this insight, we find this consistency has surprising implications for complex reasoning. Aligned Chain-of-Thought (CoT) models (e.g., DeepSeek-distilled models), for instance, leverage this effect; by generating longer reasoning chains, they push generation into later, more deterministic (lower BF) stages, resulting in more stable outputs. We hypothesize that alignment tuning does not fundamentally change a model's behavior, but instead steers it toward stylistic tokens (e.g., "Sure") that unlock low-entropy trajectories already present in the base model. This view is supported by nudging experiments, which show prompting base models with such tokens can similarly reduce BF. Together, our findings establish the BF framework as a powerful diagnostic lens for understanding and controlling LLM outputs - clarifying how alignment reduces variability, how CoT promotes stable generations, and how base models can be steered away from diversity.

## 1 Introduction

While alignment tuning improves helpfulness and safety in large language models (LLMs), it often introduces a trade-off: reduced output diversity (Padmakumar & He, 2024; Chakrabarty et al., 2024; Tian et al., 2024; Kirk et al., 2024; Lu et al., 2025) and increased determinism (Saparov & He, 2023; Song et al., 2024; Renze & Guven, 2024; Bigelow et al., 2024; West & Potts, 2025). As a result, aligned models are frequently observed to be less sensitive to different decoding strategies—a phenomenon we confirm in our own case study (§ 5.1). Similarly, Chain-of-Thought (CoT) prompting (Wei et al., 2022), while enhancing reasoning, often reduces the variance of answers. These observations point toward a common underlying phenomenon: *LLM Probability Concentration* (Figure 1a), where the model's vast potential output space collapses into a narrow set of likely trajectories.

But how should we rigorously conceptualize and measure this concentration? Autoregressive generation is inherently a traversal through a branching tree (Figure 1b). Several candidate measures exist, each capturing a different facet. However, "model perplexity" (Jurafsky & Martin, 2025) measures fit to a reference dataset rather than the model's own generative breadth, and surface-level diversity metrics (e.g., n-gram diversity (Li et al., 2016)) are often confounded by vocabulary size and output length.

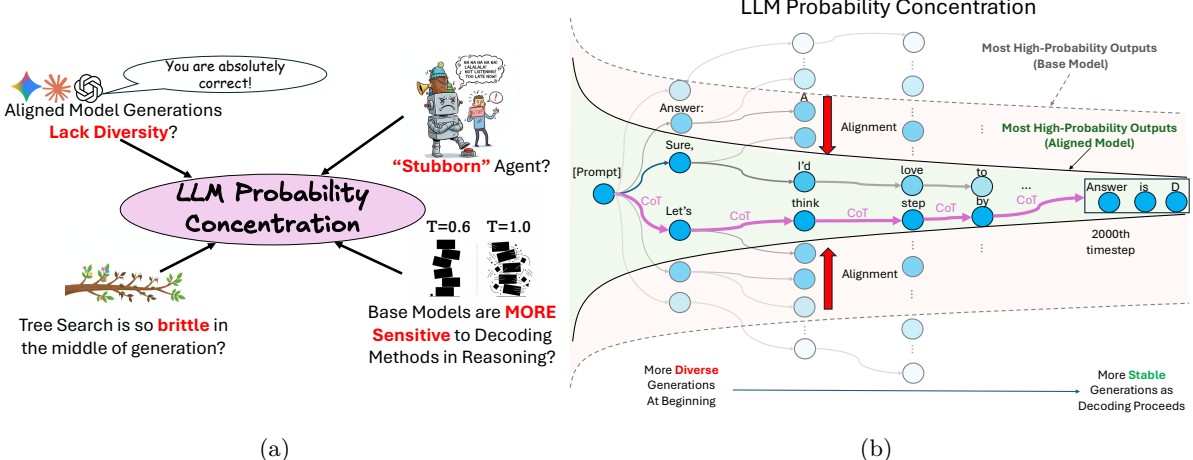

(a)                  (b)

Figure 1: (a): LLM probability concentration connects and explains several disparate yet critical phenomena in aligned LLMs. (b): A conceptual illustration of how alignment and CoT influence the generation space of LLMs. While base models begin with high output diversity, alignment tuning sharply concentrates early probability mass, leading to more stable outputs. CoT extends this effect into later positions, flattening output sample variation and reducing sensitivity to decoding.

To address this, we operationalize a principled distributional summary from information theory: the length-averaged entropy, or *entropy rate*, of the model's output distribution. We adopt the exponentiated entropy rate under the interpretive name **Branching Factor (BF)**. This metric quantifies the effective number of viable next-token choices available on average, providing a concrete, "microscopic" lens on the tree's expansion rate. Crucially, we do not propose BF as a novel mathematical metric; rather, we introduce it as a unifying conceptual framework to quantify LLM probability concentration.

Computing the entropy rate exactly is intractable in the long-horizon regime. However, by leveraging the asymptotic closeness between length-averaged log-probability and length-averaged realized entropy along typical sequences (Mudireddy et al., 2024), we can estimate BF efficiently from the model's own naturally sampled outputs, avoiding the need for teacher forcing or exhaustive enumeration.

The core contribution of this work is using the BF framework to provide a *unified explanation* for disparate LLM behaviors. We show that seemingly unconnected phenomena, reduced diversity in alignment, decoding insensitivity, and CoT stability, are all manifestations of probability concentration dynamics. Specifically:

① **Alignment Constrains the Branching Space:** We find that alignment tuning (e.g., RLHF) significantly reduces BF, typically by a factor of 2–5 overall, and up to an order of magnitude (e.g., $12 \to 1.2$) at the beginning positions, depending on alignment intensity. This reduction provides a quantitative mechanism for why aligned models are so insensitive to decoding parameters: there are simply fewer viable branches to prune.

② **Dynamic Concentration and CoT Stability:** We observe that BF typically declines over the course of generation, indicating that models "commit" to narrower trajectories as they generate. This explains the stabilizing effect of Chain-of-Thought: by encouraging long reasoning chains, CoT pushes the critical answer generation into later, lower-BF regions of the tree, resulting in more deterministic outcomes.

③ **Alignment Surfaces Latent Low-Entropy Paths:** Finally, we investigate *how* alignment achieves this concentration. Through "nudging" experiments, we show that conditioning a base model on a short, aligned-style prefix is sufficient to trigger a rapid drop in BF. This suggests that alignment does not fundamentally reshape the model's manifold but rather steers generation toward low-entropy subspaces that are already latent in the pre-trained model.

In summary, by viewing generation through the lens of BF, we move beyond reporting *that* aligned models are less diverse, to explaining *how* this concentration emerges from the underlying probabilistic structure.

## 2 Background

**Autoregressive Language Models.** LLMs are typically trained to predict the next token and the probability of output $P(y_{1:N}|x;\theta)$ can be decomposed as: $P(y_{1:N}|x;\theta) = \Pi_{t=1}^{N}P(y_t|[x, y_{1:t-1}];\theta)$, where $y_{1:t-1}$ is the output up to position $t-1$, $\theta$ is the model parameter, and $x$ is the prompt (treated as fixed). Each output sample is generated via token-by-token sampling, and the generation of multiple samples naturally forms a search tree (Yao et al., 2023; Hao et al., 2023; Wan et al., 2024). Modern LLMs go through multiple training stages. In this paper, we would use *base models* to refer to the models trained without *alignment tuning* techniques (Touvron et al., 2023), including instruction tuning and Reinforcement Learning from Human Feedback (RLHF) (Ouyang et al., 2022; Bai et al., 2022) (e.g., "Llama-2-13B" (Touvron et al., 2023)) and *aligned models* to refer to models undergoing these additional fine-tuning stages (e.g., "Llama-2-13B-Chat").

**LLM Decoding and Entropy.** Though LLMs are trained with a large vocabulary size $|V|$, the desired tokens often concentrate on a much smaller set of tokens under distribution $P(y_t|x, y_{1:t-1};\theta)$. Common decoding methods (Holtzman et al., 2020; Hewitt et al., 2022) utilize this observation and propose various heuristics to truncate vocabulary $V$ as $V_t$ at each step $t$. The next token is then sampled from the renormalized distribution $\tilde{P}(y_t|[x, y_{1:t-1}];\theta) = \mathbb{1}(y_t \in V_t)\frac{P(y_t|x, y_{1:t-1};\theta)}{\sum_{y_t \in V_t} P(y_t|x, y_{1:t-1};\theta)}$. Since tokens are sampled from the truncated distribution $\tilde{P}$,[1] we use $\tilde{P}$ to compute the empirical (token-level) entropy $\tilde{H}$ for a given prefix instance $y_{1:t-1}$:[2]

$$\tilde{H}(Y_t|[x, y_{1:t-1}];\theta) = -\sum_{y_t} \tilde{P}(y_t|[x, y_{1:t-1}];\theta) \log \tilde{P}(y_t|[x, y_{1:t-1}];\theta) \tag{1}$$

Note that $\tilde{H}$ is a *random variable* that depends on the specific realization of the prefix sequence $Y_{1:t-1} = y_{1:t-1}$. The more common notion of *conditional entropy* is thus the expectation of $\tilde{H}$ over all possible $y_{1:t-1}$:

$$\tilde{H}(Y_t|[x, Y_{1:t-1}];\theta) = \mathbb{E}_{y_{1:t-1}}\tilde{H}(Y_t|[x, y_{1:t-1}];\theta) \tag{2}$$

Conventionally, we use uppercase $Y$ to denote the *random variable* for an output and lowercase $y$ for its specific realization. Finally, along a realized sequence $y_{1:t-1}$, we define the *realized entropy* as[3]

$$h_{\text{realized}}(y_{1:N}) \stackrel{\text{def}}{=} \sum_{t=1}^{N} \tilde{H}(Y_t|y_{1:t-1};\theta). \tag{3}$$

This metric and its sequence-level reductions (e.g., mean pooling) are of profound practical importance: while estimating the full conditional entropy $\tilde{H}(Y_t|x; Y_{<t})$ requires marginalizing over an exponential space of prefix trajectories, $h_{\text{realized}}$ serves as the widely-adopted tractable proxy for quantifying generation uncertainty (Kuhn et al., 2023; Farquhar et al., 2024), generation diversity and creativity (Dušek et al., 2020; West & Potts, 2025) and controlling exploration in RLVR (Cheng et al., 2025; Cui et al., 2025; Wang et al., 2025a).[4] Linearity of expectation and the chain rule for entropy then gives us

$$\mathbb{E}_{y_{1:N}}[h_{\text{realized}}(y_{1:N})] = \sum_{t=1}^{N} \tilde{H}(Y_t|[x, Y_{1:t-1}];\theta) = \tilde{H}(Y_{1:N}|x;\theta), \tag{4}$$

i.e. $h_{\text{realized}}$ is an unbiased estimator of the marginal entropy of the whole generative process.

**A Note on Practical Text Generation.** Throughout this paper, we model an LLM as generating sequences of a fixed maximum length $N$, with realizations denoted by $y_{1:N}$, and some theoretical results consider the asymptotic regime $N \to \infty$. In practice, however, generation often terminates early, producing a shorter sequence $y_{1:n}$ with $n < N$. Our framework accommodates this by defining an *equivalent sequence* $\hat{y}_{1:N}$ such that $\hat{y}_t = y_t$ for $t \leq n$, and $\hat{y}_{n+1:N}$ consists of repeated special tokens (e.g., EOS) indicating termination. We further assume $\tilde{P}(Y_{n+1:N} = \hat{y}_{n+1:N} \mid [x, y_{1:n}];\theta) = 1$, which can be viewed as a property of pretrained LLMs. Under this convention, the probability of the equivalent sequence satisfies $\tilde{P}(\hat{y}_{1:N} \mid x;\theta) = \tilde{P}(y_{1:n} \mid x;\theta)$. Consequently, without loss of generality, we may treat all generations as having length $N$.

---

[1]Our main experiments employ mild decoding settings ($T$=1.0, $p$=0.9). These settings approximate the full distribution, align with standard evaluation practices, and ensure coherent generation from base models. Stronger truncation settings are explicitly noted where applied.

[2]The common convention setting $0 \log 0 = 0$ for entropy computation is followed.

[3]For notational brevity, we hereafter omit explicit conditioning on the input $x$ when the context is clear.

[4]See more related work discussions in § 7

## 3 Measuring LLM Branching Factor

**Probability Concentration and the Branching Factor.** The generative process of language models can be viewed as moving down a branching tree, with each token choice selecting a path forward. While the theoretical search space spans $O(|V|^N)$ sequences for vocabulary size $|V|$ and fixed sequence length $N$, LLMs concentrate the vast majority of probability mass on a much smaller subset of trajectories (Holtzman et al., 2020; Hewitt et al., 2022). This high-probability subset forms a complex, sparse "effective tree" $\mathcal{T}$.

To quantify the size of this effective tree, we utilize the concept of exponentiated entropy (perplexity). We define the effective set size $|\mathcal{T}|$ as:

$$|\mathcal{T}| \stackrel{\text{def}}{=} \exp\left(\tilde{H}(Y_{1:N}|x;\theta)\right). \tag{5}$$

Information-theoretically, $|\mathcal{T}|$ reflects the size of a uniform distribution (a "fair die") that would possess the same total uncertainty (entropy) as the model's actual complex distribution over sequences of length $N$ (O'Connor, 2013).

**Defining Branching Factor via Balanced Tree Model.** Because the exact topology of the effective tree $\mathcal{T}$ is irregular and intractable, we map it to an *equivalent balanced B-ary tree* of the same depth $N$. A perfectly balanced tree with constant branching factor $B$ and depth $N$ contains $B^N$ leaf nodes. By equating this theoretical leaf count to the effective set size ($B^N = |\mathcal{T}|$), we derive the **Branching Factor (BF)** as the geometric mean of the branching width:

$$B \equiv B(x;\theta) \stackrel{\text{def}}{=} |\mathcal{T}|^{1/N} = \exp\left(\frac{1}{N}\tilde{H}(Y_{1:N}|x;\theta)\right) = \exp\left(\bar{H}(Y_{1:N}|x;\theta)\right) \tag{6}$$

where $\bar{H}(Y_{1:N}|x;\theta)$ denotes the length-averaged marginal entropy of the sequence. $B(x;\theta)$ thus provides a normalized, token-invariant metric: it quantifies the effective number of plausible next-token choices available to the model on average at any given step.

**Linking Entropy and Log-Likelihood in Long Sequences.** Calculating the exact Branching Factor requires the total entropy $\tilde{H}(Y_{1:N}|x;\theta)$. As discussed in § 2, computing this quantity directly is intractable due to the exponential number of possible trajectories. A standard Monte Carlo approach uses the *realized entropy* $h_{\text{realized}}(y_{1:N})$ of sampled sequences as a proxy. Since $h_{\text{realized}}$ is an unbiased estimator (Eq. 4), averaging it over sufficient samples converges to the true total entropy.

However, for long sequences, even calculating $h_{\text{realized}}$ becomes computationally prohibitive. It requires a full summation over the vocabulary $V$ at every generation step to compute the local entropy, incurring a total cost of $O(N \cdot |V|)$. In contrast, computing the sequence's negative log-likelihood (NLL) involves only the probabilities of the selected tokens, scaling linearly as $O(N)$. To enable efficient estimation for long horizons, we effectively need a second level of approximation: using the computationally cheap NLL as a proxy for the realized entropy.

Standard Asymptotic Equipartition Property (AEP) theory (Shannon, 1948) suggests that for stationary processes, the length-averaged NLL of a typical sequence will converge to the (length-averaged) total entropy. However, LLM generation is neither stationary nor ergodic. Fortunately, Mudireddy et al. (2024) demonstrate that a robust connection persists without these strict assumptions: the NLL converges to the *realized entropy* $h_{\text{realized}}$ instead. We characterize this relationship as follows:[5]

**Theorem 3.1 (Log-Likelihood Convergence for LLMs)** *Given $0 < \epsilon < 1$, as $N \to \infty$:*

$$P\left(\left|-\frac{1}{N}\log\tilde{P}(y_{1:N}|x;\theta) - \frac{1}{N}h_{realized}(y_{1:N})\right| < \epsilon\right) \to 1 \tag{7}$$

Since $\mathbb{E}[h_{\text{realized}}] = \tilde{H}(Y_{1:N}|x;\theta)$, this theorem justifies using the NLL of sampled sequences as a *proxy* for realized entropy in long horizons, provided the variance is low. As an empirical verification for Theorem 3.1, we plot NLL and realized entropy for sampled outputs of Llama-3-8B-Instruct over multiple datasets[6] in

---

[5]We provide a simplified proof in § H with minor changes to the original proof of Mudireddy et al. (2024). Notably, our goal is only to show the approximation between length-averaged log-likelihood and entropy for a typical sequence, so we do not require stricter assumptions like ergodicity or stationarity.

[6]For dataset-specific details, we refer readers to § B.

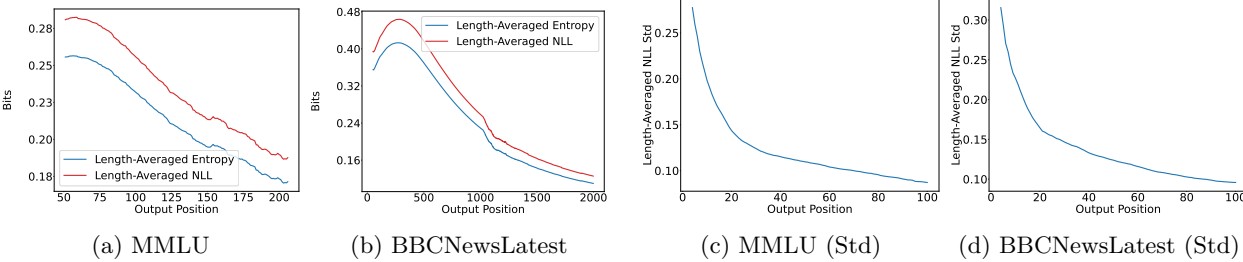

(a) MMLU     (b) BBCNewsLatest     (c) MMLU (Std)     (d) BBCNewsLatest (Std)

Figure 2: **Convergence of NLL and Entropy.** (**a, b**): The length-averaged NLL closely tracks the length-averaged Entropy. (**c, d**): The standard deviation of the length-averaged NLL diminishes rapidly with output length.

Figure 2. We observe that: as output length increases, the deviation between NLL and realized entropy vanishes, and the standard deviation of the estimator diminishes rapidly (within the first 10 tokens).

**Empirical Estimator for Branching Factor.** We now translate the theoretical definition of $B(x;\theta)$ into a practical estimator $\tilde{B}(x;\theta)$. This requires addressing two practical realities: (1) we rely on finite Monte-Carlo samples rather than full distribution access, and (2) while our theory assumes a fixed $N$, generation in practice terminates dynamically upon emitting an EOS token.

We estimate BF using $M$ independent sampled sequences $y^{(1)}, \dots, y^{(M)}$. To handle the computational constraints described above, we adopt a *hybrid estimator*. For short sequences, we compute the exact realized entropy $\tilde{H}(Y_{1:|y|}|x;\theta)$ at every step (since the vocabulary is truncated to $V_t$, this is tractable). To avoid the GPU memory bottleneck of computing the full distribution for long sequences, we approximate entropy using NLL, a substitution justified by Theorem 3.1. We explicitly avoid a naive Monte-Carlo entropy estimator on long sequences (i.e., averaging $-\log \tilde{P}(y_t^{(i)} \mid [x, y_{<t}^{(i)}]; \theta)$ over the sampled tokens *without* the inner full-distribution sum), which systematically *underestimates* entropy at finite sample budgets because it misses the long tail of the distribution – a caveat we quantify in § C. The AEP-justified NLL proxy sidesteps this issue by providing a sample-efficient surrogate for the realized entropy.

We define the estimator $\tilde{B}(x;\theta)$ by averaging over the sampled trajectories:

$$\tilde{B}(x;\theta) \approx \exp\left(\frac{1}{M}\sum_{i=1}^{M}\mathcal{E}(y^{(i)})\right), \quad \mathcal{E}(y) = \begin{cases} \frac{1}{|y|}h_{\text{realized}}(Y_{1:|y|}|x;\theta) & \text{if } |y| < L_\tau \\ -\frac{1}{|y|}\log \tilde{P}(y|x;\theta) & \text{otherwise} \end{cases} \tag{8}$$

where $|y|$ is the realized length of the sample (up to EOS) and $L_\tau$ is a threshold length. Note that explicitly normalizing by the realized length $|y|$ adapts our fixed-$N$ theory to variable-length practice, effectively measuring the branching rate per *active* generation step.

Finally, to obtain the task-level Branching Factor, we average over the dataset $X$: $\tilde{B}(X;\theta) = \sum_x p(x)\tilde{B}(x;\theta)$. Unless otherwise specified, BF refers to this dataset-level branching factor in the following sections.

## 4 Benchmarking and Attributing Branch Factors

**Models and Sampling.** We run experiments on models from Llama-2 (Touvron et al., 2023) and Llama-3 (Dubey et al., 2024) families as they are widely-used open-weight model families. For each model family, we include both base and aligned models to investigate how alignment tuning affects BF. We set $p{=}0.9$ and $T{=}1.0$ to sample outputs to conform with the setting for most datasets.

We set $M{=}50$ sequences to estimate BF, which yields a reliable estimation across datasets in prior studies. For aligned models, we apply the official chat templates to prompts. In addition, we carefully control the lengths of all inputs plus outputs to be within the context window of the models.

**Tasks.** We consider a variety of tasks covering common application scenarios of LLM generation, including reasoning and open-ended generation: MMLU (Hendrycks et al., 2021) (Reasoning), COGNAC (Chen et al., 2022) (Controlled Generation), BBCLATESTNEWS (Li et al., 2024b) (News Generation), and CREATIVE

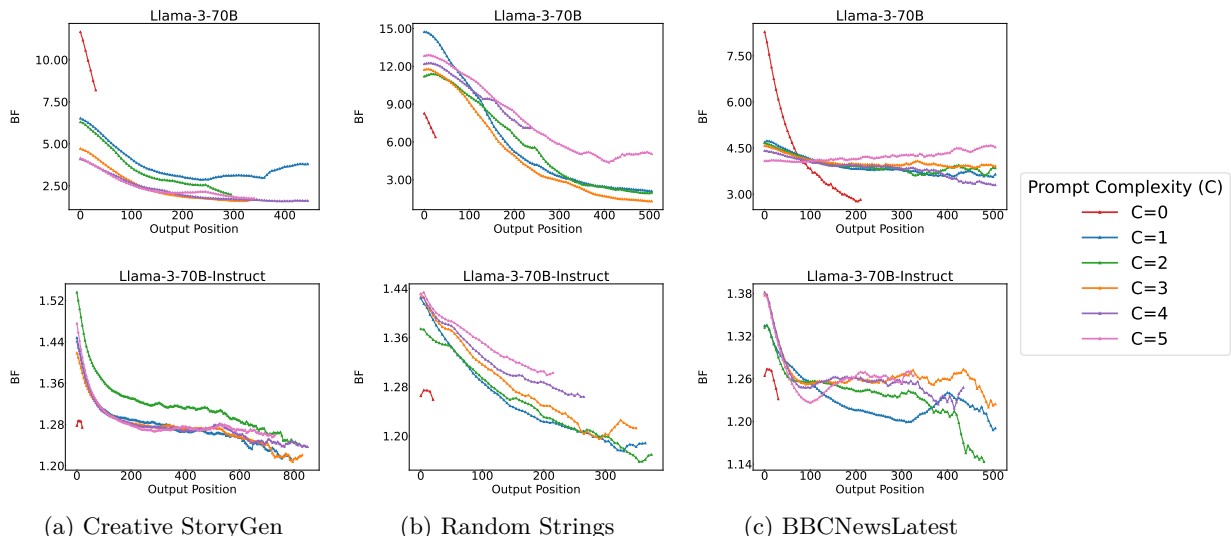

(a) Creative StoryGen  (b) Random Strings  (c) BBCNewsLatest

Figure 3: **Shrinking BF with output length over various tasks for Llama-3-70B and Llama-3-70B-Instruct.** For better visualization, we compute the exponential moving averaged values of BF with the smoothing factor set as 0.1.

STORYGEN (Chakrabarty et al., 2024) (Creative Generation). To test subjective randomness bias (Bigelow et al., 2024), we also prepare a synthetic task RANDOM STRINGS where the prompt is generated via random characters. See § B for dataset details.

**Impact Factors (IFs).** We consider modulating these factors that may impact BF computations: PROMPT COMPLEXITY ($C$), ALIGNMENT TUNING ($AT \in \{\text{Instruct}, \text{Base}\}$), MODEL SIZE ($S \in \{8\text{B}/13\text{B}, 70\text{B}\}$), and MODEL GENERATION ($G \in \{\text{Llama-2, Llama-3}\}$). $C$ controls the informativeness of the input prompt $x$ (e.g., the number of banned words in Cognac, the number of in-context samples in MMLU). Intuitively, providing more information in $x$ should make the model more confident in its outputs, resulting in a lower BF. Dataset-specific setups for $C$ are detailed in § B. $AT, S, G$ represent model-wise variations to explore how different configurations of $\theta$ affect $B(\text{X}; \theta)$.

### 4.1 BF Dynamic in Generation Process

Both BF and the output length $N$ are functions of the output Y, and BF computation relies on $N$. To avoid confounding effects, we first analyze how BF varies with $N$ before intervening IFs. In Figure 3, we demonstrate BF trajectories over different output positions by running Llama-3-70B and Llama-3-70B-Instruct on three representative tasks. Specifically, we compute BF over every five output tokens, conditioning on the prompt and all previously generated output tokens.[7] Our findings also generalize to summarization, multilingual tasks, OLMo-2 (OLMo et al., 2024) /Qwen family (Qwen, 2025) (§ G).

As we can see, first, **the base model's BF is often significantly higher than the aligned models, roughly 2–5 times**. The nearly order-of-magnitude difference is also a frequent pattern in strongly aligned models when we compare base and aligned models at the beginning of outputs. Therefore, there are actually very few candidate next-token to be truncated in decoding for the aligned models. This explains why the decoding methods exert weaker effects for aligned models (Song et al., 2024; Renze & Guven, 2024; Shi et al., 2024a), as we will see in § 5.1. Also, in most cases, **BF would often drop smoothly as more output tokens are generated**. Under the same task, when $C > 0$, different $C$ mainly controls the starting point and the rate of decreasing, while in the end, they would converge to roughly the same point. When almost zero knowledge is provided ($C = 0$), the output will end much earlier compared to $C > 0$ cases. These findings also provide support that the future token generation is gradually becoming predictable and the model may have a certain generation plan to follow, resonating with recent observations in interpretability (Pal et al., 2023; Wu et al., 2024; Li et al., 2024a) and inference acceleration (Cai et al., 2024; Welleck et al., 2024).

---

[7]See § D for full results across all models and tasks.

We emphasize that this downward trend should be read as a robust aggregate tendency of autoregressive self-conditioning, not a monotone token-wise theorem: it appears even for RANDOM STRINGS, where no semantic structure is available. Alignment separately lowers the BF level and often steepens the decline. We separate these two effects with a controlled prefix-substitution intervention in § L.

**A Matched-Length Control for CoT.** A natural concern is that CoT generations are longer, so their lower BF might simply reflect the general decline of BF over output position. To control for this, we compare BF at fixed truncated lengths on MMLU between reasoning-oriented models (DeepSeek-R1-Distill-Llama-8B/70B) and direct-answer instruct models (Llama-3-8B/70B-Instruct), following the same measurement protocol as above. Each point on the x-axis corresponds to BF computed up to the same truncated length, enabling a fair matched-position comparison.

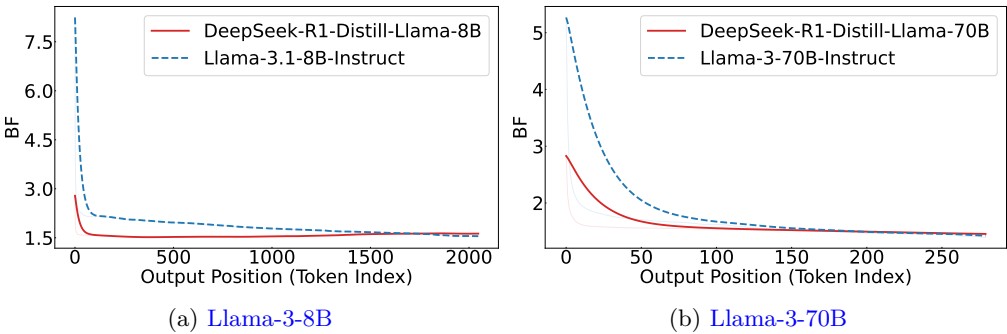

(a) Llama-3-8B  (b) Llama-3-70B

Figure 4: **Matched-length BF comparison on MMLU.** At every truncated length, the reasoning-oriented models (DeepSeek-R1-Distill-Llama) exhibit lower BF than the direct-answer instruct models, including at the earliest matched positions.

As shown in Figure 4, the reasoning-oriented models exhibit consistently lower BF across all truncated positions, including the earliest generation stages. This suggests that their lower BF is not merely a byproduct of longer generations drifting into lower-BF regions, and provides a practical check that BF comparisons are robust to this length confound. We further examine potential confounds such as prompt likelihood and data contamination in § N, and find they do not fully account for the observed BF reductions.

## 4.2 Pareto Analysis of BF

How dominant is this alignment effect compared to other factors? Figure 5a offers a first look: the Base/Aligned BF ratio is consistently high ($\gg 1$) across diverse tasks and models, peaking at $10\times$ for RANDOM STRINGS (task-wide) and appearing overall mildest for OLMo-2 (model-wide). To rigorously rank alignment against model size ($S$), generation ($G$), and prompt complexity ($C$), we perform a Pareto analysis (Figure 5) for Llama models. For each factor $D_i$, we define the unnormalized *Impact* $\tilde{I}(D_i)$ as the average absolute pairwise difference in BF when varying $D_i$ while holding other dimensions constant:

$$\tilde{I}(D_i) = \frac{\sum_{d_i,d_j \in \text{Domain}(D_i), d_i \neq d_j} |\text{Avg}(\text{B}(\cdot|D_i = d_i)) - \text{Avg}(\text{B}(\cdot|D_i = d_j))|}{|\text{Domain}(D_i)| \times |\text{Domain}(D_i) - 1|}. \tag{9}$$

Then we normalize it as $I(D_i) = \frac{\tilde{I}(D_i)}{\sum_k \tilde{I}(D_k)}$.

The results crisply validate our intuition: **alignment tuning is the primary driver of BF reduction**. Across all tasks, it consistently crosses or approaches the 80% cumulative impact threshold, surpassing all other factors by a large margin. This is consistent with the cross-sample diversity reduction independently documented by Kirk et al. (2024) and Padmakumar & He (2024); BF re-expresses the same phenomenon at the distributional, per-step level, which is what lets us connect it to decoding/inference-time behaviors (§ 5) and decompose it into stage-wise contributions (§ 6).

Among the secondary factors, for tasks with richer inputs–such as MMLU (with more in-context examples) and BBCLATESTNEWS (with more headlines)–prompt complexity $C$ and model size $S$ emerge as the next

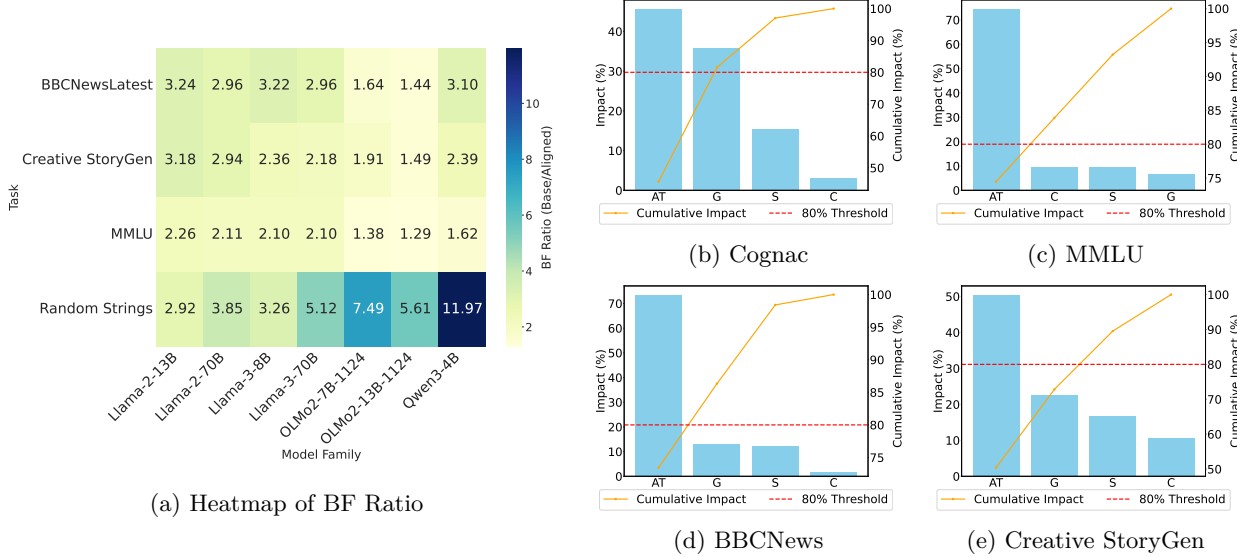

(a) Heatmap of BF Ratio

(b) Cognac

(c) MMLU

(d) BBCNews

(e) Creative StoryGen

Figure 5: **Attributing BF Reduction.** (a) Heatmap of average BF ratio (Base/Aligned) across tasks and models. The numbers indicate the average ratio over all constraint levels. Note that for OLMo-2, we follow the convention in (OLMo et al., 2024) and treat the DPO version as the aligned model. Since both base and aligned models in each column share the same tokenizer, the within-family tokenizer scaling cancels in the ratio; cross-family ratio magnitudes should be interpreted as patterns rather than precise comparisons. (b)-(e) Pareto Analysis of BF across various IFs. *AT* indicates whether the model is aligned. *C* denotes the prompt complexity. *S* refers to model size, and *G* refers to model generation.

most impactful. Prompt complexity $C$ has a noteworthy effect: contrary to intuition, more context provided in the prompt does not always reduce BF but can in fact increase it, potentially due to the cognitive burden of processing complex linguistic structures such as negated constraints in Cognac. A detailed case study and comprehensive task-wise BF results are presented in §§ E and F. In contrast, for open-ended tasks like Cognac and Story Generation, model generation $G$ plays a more dominant role, particularly improvements from Llama-2 to Llama-3. This shift likely reflects gains from the use of larger, more diverse datasets in training (Dubey et al., 2024).

# 5 Low Branching Factor Explains Generative Stability and Commitment

Our analysis in § 4 established that BF declines over the generation process (§ 4.1) and is significantly lower in aligned models (§ 4.2). This structural concentration of probability mass provides a unified probabilistic explanation for three distinct generative behaviors: the insensitivity of aligned models to decoding hyperparameters, their reduced variance in majority voting, and the high performance cost of late-stage exploration.

## 5.1 Insensitivity to Decoding Configurations

Model developers adopt different decoding strategies when reporting LLM capabilities (Touvron et al., 2023; Dubey et al., 2024; Yang et al., 2024; Guo et al., 2025). The effectiveness of strategies like nucleus sampling ($p$) and temperature sampling ($T$) relies on the assumption that a diverse set of plausible next tokens exists. However, our BF analysis suggests that for aligned models, the "plausible set" is much smaller. If this holds, we expect aligned models to be robust to decoding choices, as there are few alternatives for the decoding algorithm to select even at higher temperatures.

We verify this by benchmarking decoding methods on MMLU-STEM (Hendrycks et al., 2021), extending prior work (Song et al., 2024; Renze & Guven, 2024; Shi et al., 2024a) to the latest models including

DeepSeek-distilled models (Guo et al., 2025), which would generate long CoT before the final answer.[8] Specifically, we evaluate model performance on MMLU-STEM (Hendrycks et al., 2021) under CoT prompting across different temperatures ($T$= 0.6/1.0) in temperature sampling and truncation thresholds ($p$=0.9/1.0) in nucleus sampling (Holtzman et al., 2020). Further implementation details can be found in § A.

As shown in Table 1, aligned models exhibit limited performance variation (typically $< 10\%$) even when shifting from greedy-like settings to high temperatures. In contrast, Base models, which maintain higher BF, show significant sensitivity (up to 31%) to decoding parameters. Notably, DeepSeek-distilled Llama-8B, which generates long CoT and thus maintains a consistently low BF throughout generation, exhibits the smallest relative performance changes among 8B models. This empirically supports that *low BF effectively nullifies the impact of sampling method choices.*

| Models | Default ($T$=0.6, $p$=0.9) | $T$=0.6, $p$=1.0 | $T$=1.0, $p$=0.9 | Min ($T$=1.0, $p$=1.0) | $\frac{\text{Default}-\text{Min}}{\text{Default}}\%$ |
|---|---|---|---|---|---|
| Llama-3-70B-Instruct | 78.50 ($\pm$ 2.09) | 77.60 ($\pm$ 2.23) | 77.50 ($\pm$ 2.60) | 75.90 ($\pm$ 2.85) | 3.31 |
| Llama-3-70B | 78.00 ($\pm$ 3.52) | 74.00 ($\pm$ 3.80) | 72.00 ($\pm$ 4.38) | 63.50 ($\pm$ 5.02) | 18.59 |
| DeepSeek-R1-Distill-Llama-8B | 66.30 ($\pm$ 3.51) | 65.70 ($\pm$ 3.84) | 62.70 ($\pm$ 4.14) | 59.70 ($\pm$ 4.65) | 9.95 |
| Llama-3.1-8B-Instruct | 63.00 ($\pm$ 4.01) | 61.50 ($\pm$ 4.37) | 57.50 ($\pm$ 4.92) | 50.50 ($\pm$ 5.34) | 19.84 |
| Llama-3.1-8B | 54.00 ($\pm$ 4.61) | 53.50 ($\pm$ 4.92) | 47.00 ($\pm$ 5.21) | 37.00 ($\pm$ 5.48) | **31.48** |

Table 1: **Experiment Results across decoding methods on STEM subset of MMLU.** We follow the common practice of using 5-shot CoT prompting. $\frac{\text{Default}-\text{Min}}{\text{Default}}\%$ indicates the maximum relative performance drop when deviating from the default decoding configuration.

| Model | Maj@1 Std | Maj@3 Std | Maj@8 Std | Maj@16 Std | BF |
|---|---|---|---|---|---|
| DeepSeek-R1-Distill-Llama-70B | **14.34** | **8.29** | **4.99** | **3.21** | **1.23** |
| Llama-3-70B-Instruct | 16.37 | 11.40 | 7.50 | 5.12 | 1.28 |
| Llama-3-70B | 27.78 | 19.53 | 13.22 | 9.23 | 1.31 |
| DeepSeek-R1-Distill-Llama-8B | **27.10** | **20.91** | **13.93** | **9.14** | **1.23** |
| Llama-3.1-8B-Instruct | 31.54 | 24.64 | 17.30 | 12.90 | 1.31 |
| Llama-3.1-8B | 36.41 | 29.78 | 20.43 | 14.05 | 1.35 |

Table 2: **Majority Voting@K standard deviation on MMLU-STEM with** 200 **samples.** We compute the standard deviation over 100 bootstrapping trials, each using 64 samples per instance. We set $T = 0.6, p = 0.9$ to match standard benchmarking settings, differing from $T = 1.0, p = 0.9$ setup in § 4. Lower temperature concentrates probability mass on fewer tokens, reducing BF and complicating direct comparisons. However, bootstrapping (100 runs) reveals minimal variability ($\approx 0.01$), confirming that the BF differences reported here remain significant. Consequently, BF remains a strong predictor of standard deviation.

## 5.2 Reduced Variance in Majority Voting

If aligned models have a restricted output space (low BF), this should also manifest as reduced variance across independent samples. Beyond simply confirming the lower sampling variance documented for instruct models in prior decoding studies (Song et al., 2024), our framework lets us *predict* that this effect should further intensify for long-CoT reasoning models in inference-time scaling: because their BF stays low throughout the long reasoning trace, the variance across independent rollouts should be even smaller than that of direct-answer instruct models at the same scale. We test this prediction by evaluating output variance on MMLU-STEM using 200 samples per model. We benchmark the standard deviation across Majority@K accuracy metrics ($K = 1, 3, 8, 16$) with $T = 0.6, p = 0.9$.

As detailed in Table 2, BF serves as a strong predictor of sampling consistency. The Long-CoT models (DeepSeek-R1-Distill), which generate significantly longer outputs and achieve the lowest global BF, consistently show the smallest performance variance. This suggests that the "stability" observed in decoding benchmarks is not merely an artifact of specific hyperparameters, but is closely tied to the narrowed generation manifold.

---

[8]For Llama-3 series models, in our prior study, we find there is only a minor performance difference between Llama-3 and Llama-3.x. We mainly use Llama-3 in this paper as it includes the most diverse model collection.

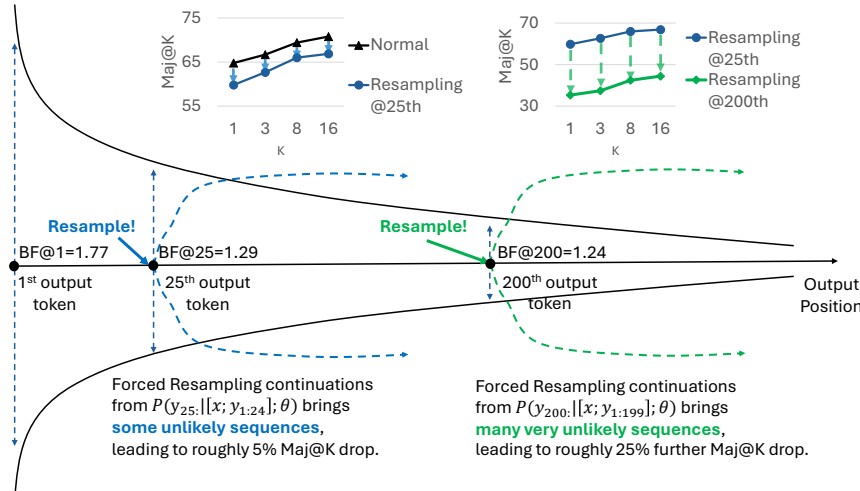

Figure 6: **Resampling from different output positions to assess the effect of interrupting BF reduction**. We resample new continuations at the 25th and 200th output token of DeepSeek-Distilled Llama-8B MMLU outputs. Results show substantial performance drops at both positions.

### 5.3 Risks of Mid-Generation Forking

The stability provided by low BF suggests a strong commitment to a specific reasoning trajectory. Does this commitment imply that the model cannot effectively explore alternative paths once generation has begun? If BF reflects a semantic "lock-in," forcing the model to branch out (fork) at late, low-BF stages should disrupt coherence and degrade performance.

To examine this, we conduct a resampling experiment using DeepSeek-Distilled Llama-8B output samples. Procedurally, for a given position $t$: 1) Take the prefix $y_{<t}$ generated by the model. 2) Sample a new continuation $y'_{\geq t}$ from $P(Y_{\geq t}|[x, y_{<t}]; \theta)$. 3) Evaluate the full sequence $[y_{<t}, y'_{\geq t}]$ on the task.

As shown in Figure 6, performance drops sharply when resampling occurs at a later, lower-BF position in the sequence. This suggests that aligned models are not just concentrating probability mass locally (reflects a "deeper commitment" to specific paths), but are actively locking into trajectories, making late-stage deviations more error-prone. In practice, this highlights a key application of BF: *parallel sampling should be applied early, while BF remains high*, to ensure meaningful diversity and avoid quality degradation.

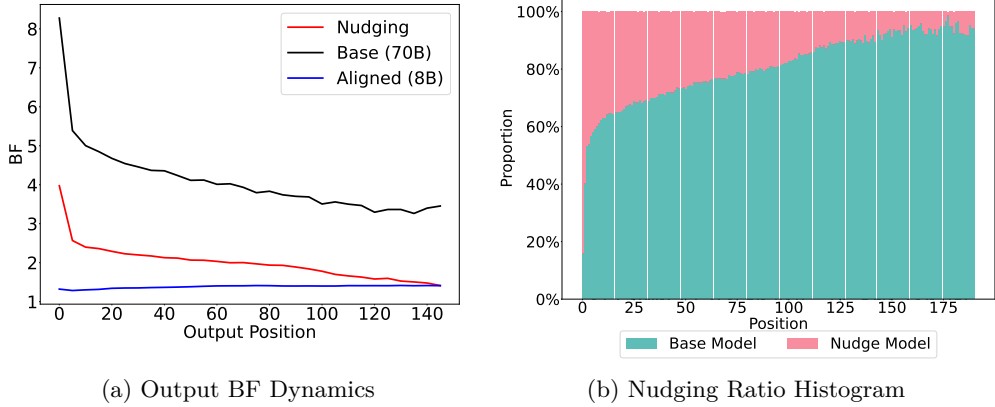

(a) Output BF Dynamics          (b) Nudging Ratio Histogram

Figure 7: Nudging Experiments over Just-Eval-Instruct.

## 6 How does Alignment Tuning Impact BF?

Why does alignment tuning exert such a pronounced effect on BF? Building on the superficial alignment hypothesis (Zhou et al., 2024) ("*Alignment tuning might simply teach base LLMs to select a subdistribution of*

*data formats for interacting with users.*") and recent tuning-free alignment work (Lin et al., 2023; Fei et al., 2024; Lake et al., 2025), we hypothesize base models already encode low-entropy conditional distributions. In this view, alignment tuning doesn't reshape generation from scratch, but instead nudge the model toward *stylistic tokens* (e.g., "Sure"), thereby narrowing the conditional distribution.

To test this hypothesis, we reproduce the nudging experiments (Fei et al., 2024), over Just-Eval-Instruct (Lin et al., 2023) and MMLU datasets. We employ Llama-3-70B for drafting most outputs. However, when the base model's Top-1 probability is low, we apply nudging by switching to Llama-3-8B-Instruct to generate a single word. Using a smaller aligned model to nudge a larger base model (70B) isolates the steering effect of the stylistic prefix itself, independent of the nudging model's raw capability. BF was computed as in prior experiments. The results, shown in Figure 7,[9] indicate that after most nudging occurs early in the generation process – indicating the prefix generated by the nudging model is of low probability. These observations collectively support our hypothesis. We emphasize the role of this experiment: Fei et al. (2024) use their nudging setup to evaluate sample-level task performance for inference-time alignment, and our goal is *not* to re-explain that observation. Instead, we adopt the same controlled setup to *test our distributional hypothesis* about how probability concentration arises during alignment – namely, that a small number of stylistic tokens is sufficient to steer a base model into the low-BF regime characteristic of aligned models. The BF measurement then provides the per-step distributional view (which prefix induces how much concentration, and where in the sequence) that sample-level metrics cannot resolve.

**Which Training Stage Reduces BF Most?** While our nudging analysis suggests alignment tuning narrows the distribution by nudging models toward stylistic tokens, it remains unclear which specific phase of the pipeline drives this reduction. Disentangling these effects is difficult because most model releases (e.g., Llama 3 (Dubey et al., 2024)) do not provide intermediate checkpoints, and modern post-training often employs iterative, interleaved schedules rather than discrete stages. However, the OLMo-2 suite (OLMo et al., 2024) releases intermediate checkpoints, enabling a preliminary stage-wise dissection. We measure BF changes across Supervised Fine-Tuning (SFT) and Direct Preference Optimization (DPO) (Rafailov et al., 2023) stages on the open-ended CREATIVE STORYGEN task. As shown in Figure 8, we observe distinct behaviors across model scales: for the 7B OLMo-2 model, SFT is the primary driver of BF reduction, causing a 42.3% drop compared with 20.5% for DPO, whereas for the 13B OLMo-2 model, the DPO stage contributes more significantly, with a 25.6% drop compared with 14.9% for the SFT stage. This divergence suggests that the impact of alignment tuning stages is not universal but sensitive to specific training recipes and model scales.

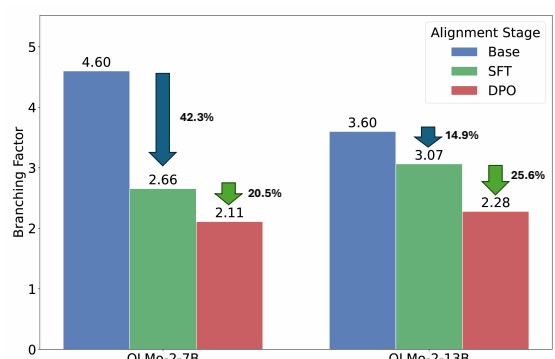

Figure 8: **Stage-wise Contribution to BF Reduction.** We analyze OLMo-2 models on the Creative StoryGen task.

**Beyond OLMo-2: Other Post-Training Algorithms.** The OLMo-2 checkpoints provide a clean stage-wise comparison, but they only cover the SFT and DPO stages. To broaden the picture, we next study a publicly released RLHF pipeline from RLHFlow (Dong et al., 2024) and OpenRLHF (Hu et al., 2024) collaboration, which includes checkpoints for SFT, DPO, PPO, and Iterative DPO variants. Because post-training recipes vary substantially across organizations (e.g., in data and training setup), we view this as a controlled case study rather than a universal ranking of algorithms. We evaluate all checkpoints on CREATIVE STORYGEN using the same setup as above.

As shown in Figure 9, SFT again accounts for the largest BF reduction (4.05 → 2.35), consistent with the OLMo-2 trend.

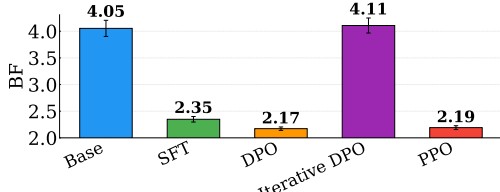

Figure 9: **BF across post-training algorithms.** Using checkpoints from an RLHFlow/OpenRLHF pipeline on Creative StoryGen, we find that SFT drives the largest BF drop, DPO and PPO yield similar further reductions, and Iterative DPO partially preserves distributional breadth.

---

[9]We present results on Just-Eval-Instruct only for brevity. MMLU results are included in § I.

DPO and PPO then produce very similar additional reductions ($2.35 \rightarrow 2.17$ for DPO and $2.35 \rightarrow 2.19$ for PPO), suggesting that these two offline preference-optimization methods concentrate the distribution to a comparable extent beyond the SFT stage. Iterative DPO, however, avoids part of this shrinkage. A plausible explanation is that its more online training loop repeatedly refreshes the reference policy and collects preference signals from the model's updated outputs, which can expose the model to underrepresented regions of the solution space instead of reinforcing a fixed offline preference set. This interpretation is consistent with Wu et al. (2025b), who argue that offline reinforcement-style methods remain tightly tethered to their initial reference policy and therefore have limited ability to move beyond it.

## 7 Related Works

**Uncertainty Quantification for LLM.** Uncertainty quantification (UQ) for LLMs has gained significant attention due to its importance in real-world applications, particularly in high-stakes domains (Desai & Durrett, 2020; Jiang et al., 2021; Wang et al., 2022; Kadavath et al., 2022; Xiong et al., 2024; Ye et al., 2024; Gupta et al., 2024). Existing methods typically address closed-domain tasks such as classification and question-answering, where outputs are discrete and easier to assess. However, as Kuhn et al. (2023) note, these approaches often overlook challenges specific to open-ended generation, such as semantic equivalence across outputs. They introduce "semantic entropy" to quantify uncertainty in LLM output space by first clustering the sampled output and then quantifying uncertainty over cluster distribution. This method empirically works well in hallucination detection (Farquhar et al., 2024). In this paper, we focus on investigating the probability concentration phenomenon for LLMs. We introduce BF to quantify this concentration, which applies broadly across tasks without imposing strong assumptions on output categories.

**Reduced Diversity in Aligned Models.** Recent studies have consistently shown that alignment tuning reduces output diversity in language models (Perez et al., 2022; Padmakumar & He, 2024; Chakrabarty et al., 2024; Tian et al., 2024; Kirk et al., 2024; Lu et al., 2025; Lake et al., 2025; West & Potts, 2025). Among these, Kirk et al. (2024) is the most closely related, with a systematic cross-sample diversity analysis under RLHF. We discuss the relationship between cross-sample diversity and BF in § K. Mechanistically, Lake et al. (2025) observe that alignment suppresses diversity by aggregating information into longer, standardized responses, though they argue this preserves useful base model behaviors. Our work builds on this inquiry by connecting reduced diversity with related observations of diminished randomness and robustness in aligned models (Saparov & He, 2023; Song et al., 2024; Renze & Guven, 2024; Bigelow et al., 2024; Shi et al., 2024a). Crucially, while previous decoding-sensitivity studies document the *symptom* (e.g., aligned models becoming insensitive to sampling hyperparameters), the BF framework pinpoints the underlying *mechanism*: the shrinking distributional landscape over the course of token-by-token generation. This conceptual lens allows us to further forecast and verify the same probability concentration effects in long-CoT models.

Traditional diversity metrics such as n-gram lexical diversity (Li et al., 2016) are sensitive to vocabulary size and output length (Liu et al., 2022; Tevet & Berant, 2021; Guo et al., 2024; Kirk et al., 2024) and cannot work well with most recent long CoT models. In § M, we demonstrate that lexical diversity poorly correlates with BF and fails to robustly measure generation concentration.

Beyond diagnostic findings, the BF lens has begun to enable concrete downstream applications, including inference-time methods for reasoning (Fu et al., 2025) and open-ended generation (Wang et al., 2025b), as well as training-time methods for reasoning (Yang et al., 2025) and joint quality–diversity optimization (Li et al., 2025).

Our work also resonates with information density research in cognitive science and linguistic theories, and we present a short discussion in § J.

## 8 Discussion

**Practical Implications.** A key practical implication of our findings is that reduced BF neglects alternative generations and forking. Consequently, simply tweaking decoding parameters (e.g., temperature), is unlikely to restore diversity without severely degrading quality (Renze & Guven, 2024). Our work offers a clear

explanation for why this occurs, particularly for methods like beam search. The resampling experiment in § 5.3 provides direct evidence that for low-BF models, off-path trajectories are not just less probable but often of lower quality. With little probability mass distributed among alternative paths, beam search has few viable options to explore, yielding diminishing returns. This suggests that efforts to mitigate diversity loss should target the training process itself – a more promising, albeit challenging, direction. Future work could involve curating more diverse alignment data or designing novel training objectives that balance instruction-following with distributional diversity (Wang et al., 2024; Kwon et al., 2024; Lanchantin et al., 2025; Chung et al., 2025). System-level interventions (e.g., model collaboration) also present a viable path forward (Fei et al., 2024; Lu et al., 2024; Venkatraman et al., 2025; Ismayilzada et al., 2025). While our paper's primary contribution is diagnostic, we believe this foundational understanding is a necessary prerequisite for developing such effective countermeasures.

That said, the above discussion applies primarily to settings where output diversity is valued, such as creative generation, open-ended dialogue, and exploratory reasoning. For tasks with unique correct answers (e.g., arithmetic, factual retrieval), low BF might in fact be desirable – it might reflect the model concentrating probability mass on the correct output, and alignment-induced BF reduction can be beneficial.

**Autoregressive Self-Narrowing vs. Alignment.** It is important to separate two effects that are easy to conflate. The *downward trend* of BF over generation is a robust aggregate tendency of autoregressive generation, not a monotonic token-wise law: as the model conditions on its own growing prefix, the next-token distribution often becomes more concentrated, even when the context is not semantically meaningful. This helps explain why BF also decreases for RANDOM STRINGS, which is otherwise puzzling. Alignment is a separate force that lowers the absolute BF level and steepens the early narrowing, rather than being the sole explanation for the decrease. We make this explicit through a controlled intervention in § L: replacing the model's own prefix with externally-sampled i.i.d. random tokens surges BF back up and then generally lets autoregression narrow it again, in both base and aligned models. A practical corollary – relevant only when diversity is desired – is that injecting out-of-distribution/unexpected content into the context can *temporarily* drag the model back to a high-BF region; but because autoregressive self-conditioning tends to re-concentrate the distribution over subsequent tokens, this is a local intervention rather than a cure. Conversely, content that is unexpected from the model's own predictive viewpoint (random strings, adversarial agentic feedback, or negation) can *raise* BF (§ L).

**Societal Homogeneity Bias of Alignment Tuning.** Our work identifies a key dynamic in modern LLMs: alignment tuning significantly reduces the Branching Factor (BF), leading to more homogenized and predictable outputs. While this can be beneficial, it also carries potential negative societal impacts. In applications such as automated content generation, creative writing, or decision-support systems (Padmakumar & He, 2024; Sorensen et al., 2024; Wu et al., 2025a; Murthy et al., 2025; Rodemann et al., 2025; Ashkinaze et al., 2025; Lake et al., 2025), this reduction in diversity could inadvertently reinforce social biases, stifle creativity, and limit the exploration of novel ideas. Rodemann et al. (2025) further argue that empirical alignment, relying on limited and potentially biased human feedback, creates selectional bias that fails to capture the full spectrum of human values. We believe that formally understanding and quantifying the mechanisms of probability concentration, as we do in this paper, is a critical and necessary first step toward developing alignment techniques that mitigate these risks and foster models that are not only helpful and harmless but also diverse and robust.

## 9 Limitations

**BF is a First-Moment Summary.** By definition, BF captures the exponentiated length-averaged entropy – a single scalar. Like any such summary, it can obscure higher-order structural properties of the generative distribution. Consider a stylized example with sequences of length $N = 2$ over vocabulary $a, b, c, d$. DISTRIBUTION A (uniform branching): $P(Y_1 = a) = P(Y_1 = b) = 1/2$, $P(Y_2 = c \mid Y_1) = P(Y_2 = d \mid Y_1) = 1/2$ regardless of $Y_1$. DISTRIBUTION B (front-loaded branching): $P(Y_1)$ is uniform over all four tokens, while $P(Y_2 \mid Y_1)$ is deterministic. Both distributions yield $\tilde{H}(Y_{1:2}) = 2 \ln 2$ and hence $BF = 2$, yet they correspond to qualitatively different trees. Nonetheless, BF is designed to answer a specific question — how concentrated

is the generation process on average – and practitioners who require finer-grained characterization of the tree topology should complement BF with positional or higher-order analyses.

**Limitations of BF Hybrid Estimator.** Our efficient BF estimator involves two finite-sample approximations, each with its own bias. First, the gap between length-averaged NLL and realized entropy shrinks with sequence length $N$ (Theorem 3.1) but can be non-negligible for short sequences. To address this, we adopt a hybrid strategy (Equation (8)): for sequences shorter than a threshold $L_\tau$, we compute the exact realized entropy via full vocabulary summation, resorting to the NLL approximation only for longer sequences where it is both accurate and computationally necessary. Figures 2(c) and 2(d) empirically validate this design: the standard deviation of the length-averaged NLL estimator diminishes rapidly, falling to practical levels within roughly 20 tokens. These observations provide concrete guidance for practitioners – exact entropy computation is feasible and recommended for short outputs (we recommend choosing a $L_\tau$ between 20 and 50), while the NLL proxy can be safely adopted beyond this early regime.

Second, the finite sample Monte-Carlo estimate of realized entropy itself underestimates the true entropy because the rare, high-entropy tail is under-sampled at finite $M$; § C characterizes this empirically (estimated BF rises monotonically as $M$ grows from 4 to 64). Absolute BF magnitudes should therefore be read as lower bounds. Cross-model and base-vs-aligned ratios are more robust to this bias than absolute magnitudes, because the under-sampling has comparable scale across models with comparable effective output spaces.

**Tokenizer Dependence and Cross-Family Comparison.** BF is computed at the token level, so its raw magnitude depends on the model's tokenizer: a coarser tokenizer (fewer, longer tokens) tends to produce smaller BF than a finer one, even for distributions of comparable conceptual breadth. Direct cross-family comparison of raw BF values is therefore not meaningful. Throughout this paper we either compare within a single family or, for cross-family analyses such as Figure 5a, report the within-family Base/Aligned BF *ratio*, which is tokenizer-invariant because each Base/Aligned pair shares a vocabulary. Practitioners adopting BF should follow the same convention.

**Surface vs. Semantic Concentration.** BF is a distributional, token-level summary and is by construction insensitive to whether two surface forms with different token sequences encode the same meaning – a distinction that semantic uncertainty methods such as Kuhn et al. (2023) are specifically designed to capture. We view these measures as complementary: BF diagnoses where probability mass is concentrated in the model's own next-token distribution, while semantic clustering diagnoses how this concentration projects onto a smaller set of meanings. Empirically, the token-level concentration captured by BF accumulates over the sequence and manifests in measurable reductions in sample-level lexical and semantic diversity for aligned models (Kirk et al., 2024; West & Potts, 2025; Lake et al., 2025); quantifying the per-prompt mapping from BF to semantic diversity is an interesting direction for follow-up work.

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

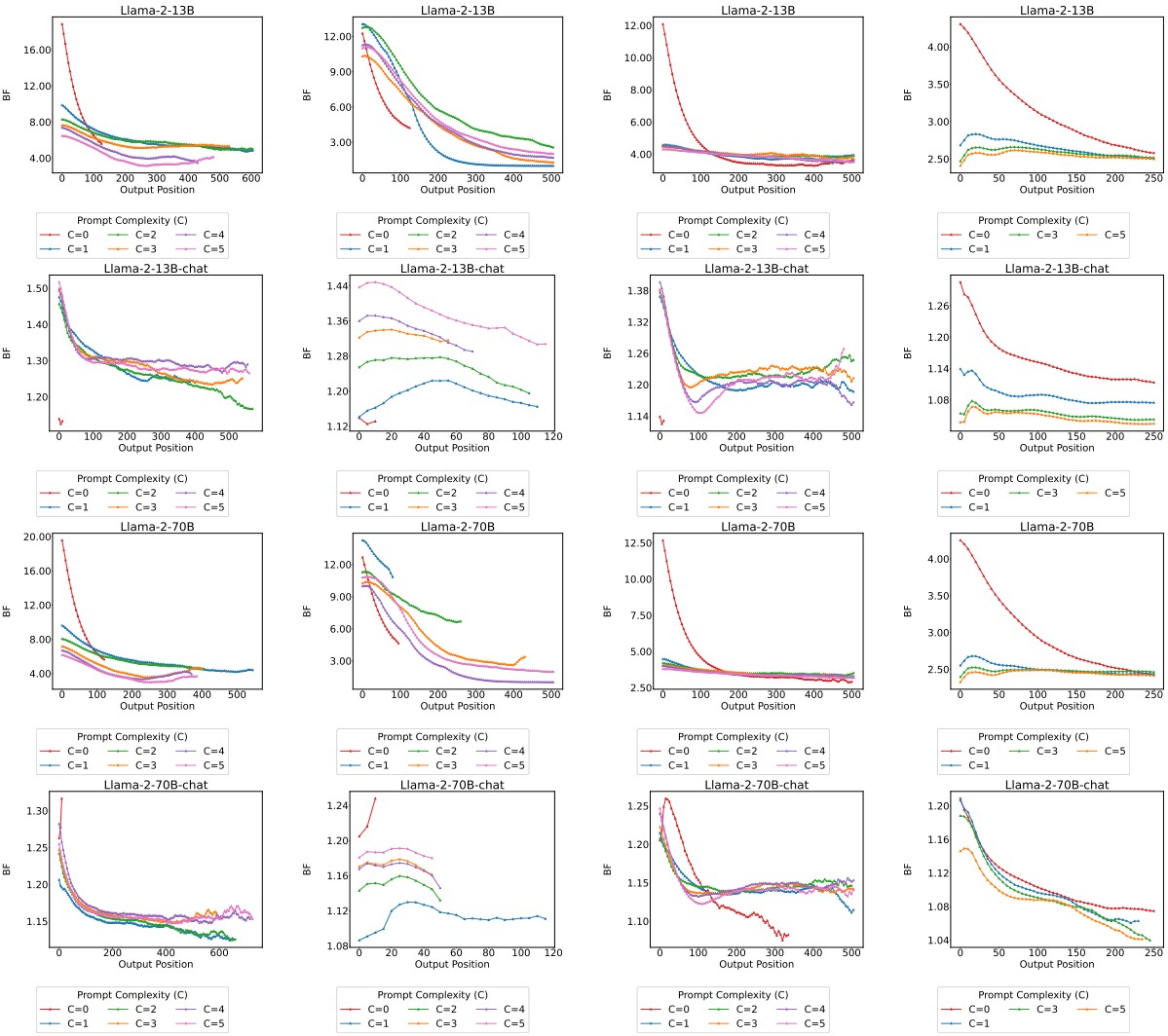

Figure 10: **BF Output Dynamic for Llama-2-families.** For better visualization, we compute the exponential moving averaged values of perplexity with the smoothing factor set as 0.1.

# A  Case Study Implementation Details

We use the scripts in Qwen-2.5-Math (Yang et al., 2024) for standard reasoning benchmarks.[10] We sample 200 examples from MMLU-STEM and compute the performance numbers under 64 trials and report the average performance.

# B  Dataset-Specific Processing

For all datasets we used in the paper, we carefully controlled whether the prompt length and expected output length would exceed the model's maximum length.

**MMLU**   (Hendrycks et al., 2021) is a widely-used multiple-choice reasoning question. Unless otherwise explained, we use the full test set of MMLU to avoid potential contamination, following benchmarking settings reported in most LLM technical reports(Touvron et al., 2023; Dubey et al., 2024; Guo et al., 2025).

---

[10]https://github.com/QwenLM/Qwen2.5-Math/tree/main

We formulate prompt complexity $C$ as the number of in-context samples. For example, $C = 1$ means we only add one in-context sample. For prompting setup and postprocessing details, we follow the standard implementation in Qwen-2.5-Math (Yang et al., 2024).

**Cognac** (Chen et al., 2022) is a controlled generation task requiring language model *not* to generate specified banned words provided in the prompt. We use the WordNet subset (Miller, 1995) of Cognac as this is the only released setting in Cognac paper, where the topic is a root node and the constraint is defined as a subtree. We sampled 200 instances using the provided data generation codes in our experiments. To ensure most model generations ended properly in the decoding process, we relax the constraint of maximum decoded tokens $T$ from 60 to 512. We use the same prompt templates following their Github repo.[11]

**Creative Story Generation** (Chakrabarty et al., 2024) provides the plots and story continuation from both machine and human. We adopt the provided 11 human-written story plots in the original dataset as the prompt. In this task, we set the maximum token $T = 1024$ to ensure the continued story written by LLM can have a proper ending. We formulate prompt complexity $C$ as providing $C \times 25$ words in the plot.

**Random Strings** Similar to Bigelow et al. (2024), we sample 200 random strings with length $L \sim U(256, 512)$ from the tokenizer vocabulary as the prompt. Prompt complexity $C$ is formulated by providing $C \times 15$ tokens in the prompt, ensuring each article contains at least 100 tokens.

**BBCLatestNews** (Li et al., 2024b) is a news collection dataset aims at collecting news that is beyond the time cut for training LLMs. Unlike creative story plots, news articles are typically more structured and organized, although headlines can still be surprising. We select news articles from January to July 2024 to minimize data contamination, as the Llama models have a knowledge cut-off in late 2023. We formulate prompt complexity $C$ as providing $C \times 15$ words in the prompt.

## C Entropy Underestimation via Monte Carlo Sampling

To demonstrate the limitations of Monte Carlo (MC) sampling for entropy estimation in long sequences, we conducted an empirical study. We prompted Llama-3-8B-Instruct with 5-shot CoT examples from the MMLU dataset. We then estimated the entropy of its generated responses using a varying number of MC samples: $M \in \{4, 8, 16, 32, 64\}$.

As illustrated in Figure 12, the estimated entropy consistently increases with the number of samples. This trend confirms that MC estimation with a small sample size systematically **underestimates** the true entropy because it fails to capture the long tail of the full probability distribution. While increasing the sample count mitigates this bias, it does so at a significant computational cost. In contrast, Theorem 3.1 allows us to use the negative log-likelihood (NLL) of a single typical sequence for a more efficient and accurate estimation.

## D Full BF Output Dynamics Investigation

Here we present full task-wise and model-wise BF output dynamic for Llama-2 in Figure 10 and Llama-3 in Figure 11. We can observe the trends as in § 4.1: ① **The average BF for the base model ($\approx 12$) is roughly ten times higher than the aligned model ($\approx 1.2$). ② BF would often drop smoothly as more output tokens are generated**.

## E Curious Case of Prompt Complexity

Intuitively, greater prompt specificity (larger $C$) reduces BF by narrowing the model's output space through more informative context. However, our experimental results reveal task-varied effects. As illustrated in Figure 13 for the Cognac task, greater prompt complexity can *increase* BF–potentially due to the cognitive burden of processing negation or complex linguistic structures. In contrast, for tasks like News

---

[11]https://github.com/princeton-nlp/Cognac/tree/main

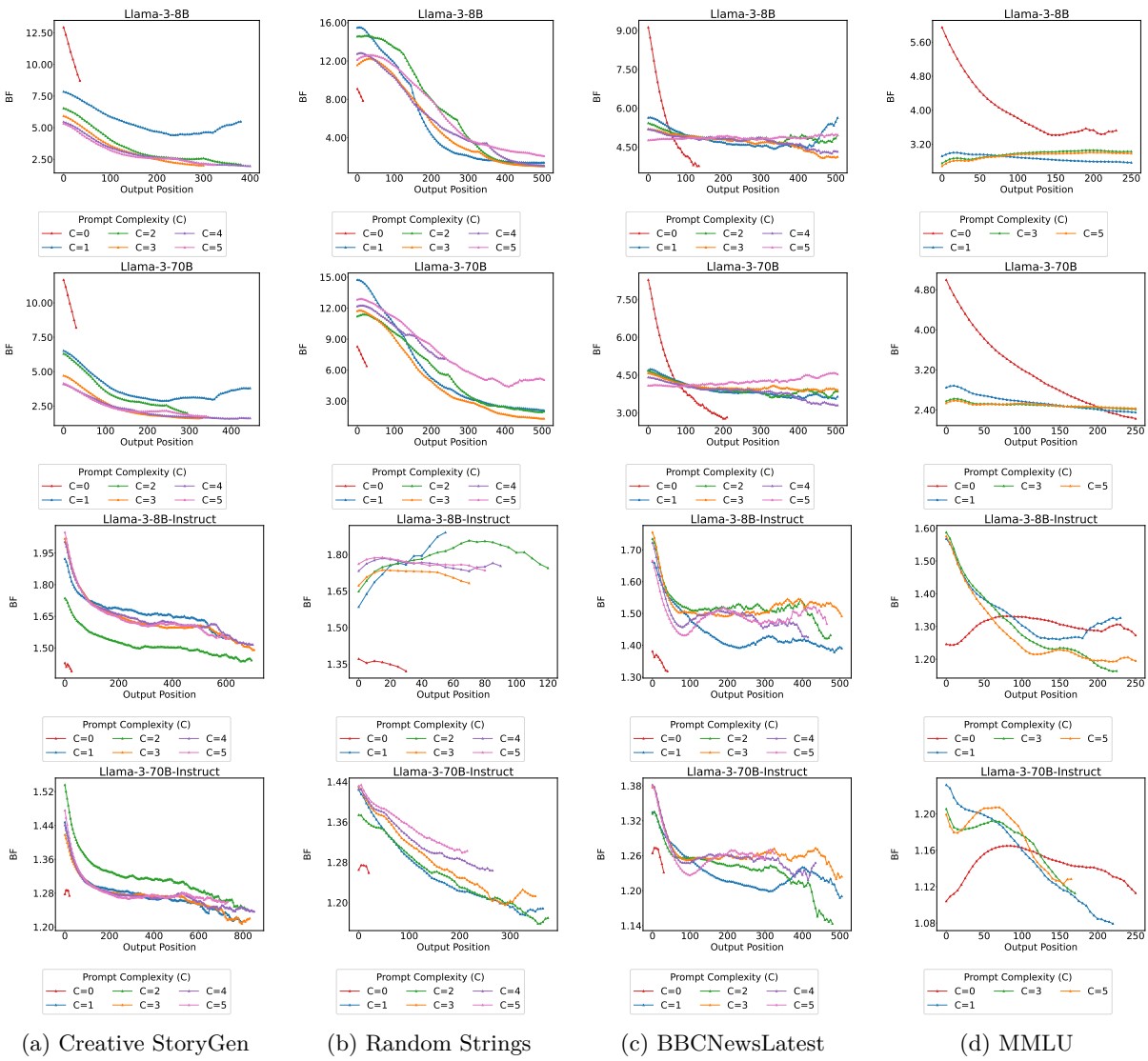

Figure 11: **BF Output Dynamic for Llama-3-families.** For better visualization, we compute the exponential moving averaged values of perplexity with the smoothing factor set as 0.1.

Generation, higher $C$ generally leads to lower BF, consistent with the expected narrowing of output diversity. Comprehensive task-wise BF results are provided in § F. This negation-induced increase is one instance of a more general pattern – content that is unexpected from the model's own predictive viewpoint raises BF – which we examine in § L.

## F    Full Task-wise BF Evaluation on Different Prompt Complexity

The full task-wise BF evaluation results over different prompt complexity can be found in Figure 14. Here we can see that prompt complexity modulates BF in highly non-consistent ways across models and tasks, and there are no clear monotonic patterns, contradicting the intuition that with more context given, the model should have more confidence in what to generate.

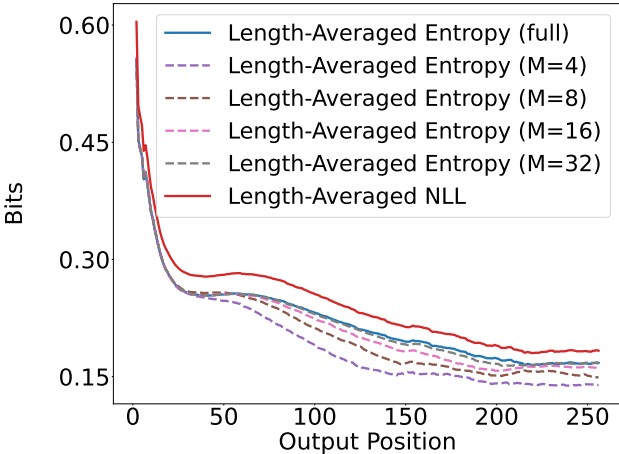

Figure 12: **Monte Carlo (MC) sampling systematically underestimates entropy.** The plot shows that the estimated entropy of sequences from Llama-3-8B-Instruct increases with the number of MC samples ($M$). A small sample size fails to cover the vast output space, leading to an underestimation of the true entropy. This bias is difficult to eliminate without incurring substantial computational costs.

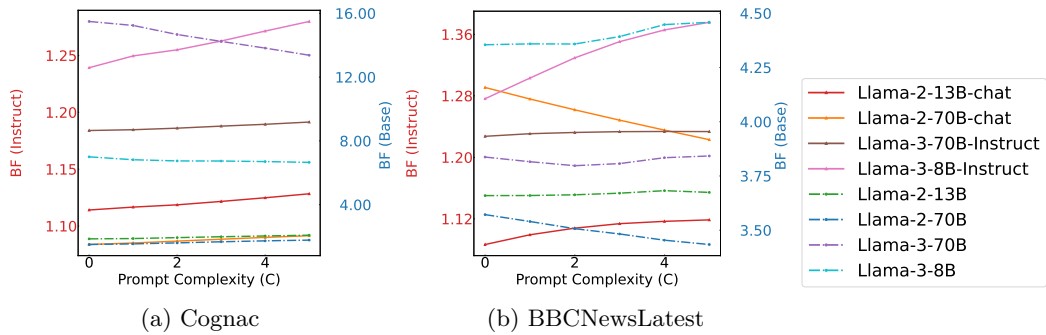

(a) Cognac                    (b) BBCNewsLatest

Figure 13: **Task-varied influence of prompt complexity $C$ on BF.** On Cognac, we see BF increases with increased $C$, while on BBCNewsLatest, increasing $C$ can lead to reduced BF.

## G   Generalization to Additional Tasks and Models

To confirm the generalizability of our findings (§ 4), we extend our experiments to new domains: summarization on XSUM (Narayan et al., 2018), multilingual tasks on AYA (Singh et al., 2024). We formulate prompt complexity $C$ as providing $C \times 25$ words in the prompt. We also verify our findings on a new model, Qwen3-4B (Qwen, 2025).[12] As presented in Figure 15, our core conclusions remain robust across these diverse conditions.

We also analyze OLMo-2 (OLMo et al., 2024) across different alignment stages (Base and DPO) on Creative StoryGen and MMLU, covering both 7B and 13B scales. As shown in Figure 16, OLMo-2 exhibits a similar trend where alignment tuning reduces BF, although the reduction is less pronounced compared to Llama models, suggesting that OLMo-2's post-training asserts less influence on the generation manifold. Additionally, we provide the BF dynamics for Qwen3-4B on MMLU in Figure 17.

## H   Proof of LLM Log-Likelihood Convergence

The following proof is a simplified version of the one in (Mudireddy et al., 2024), presented for completeness and to refine its original bounds. For the formal measure-theoretical treatment, we refer readers to the

---

[12]For the Qwen3 family, we use the Qwen3-4B-Base and Qwen3-4B-Instruct-2507 pair. Other aligned variants can be activated into a reasoning mode, exhibiting behavior distinct from the models in our main study, and were thus excluded for a fair comparison.

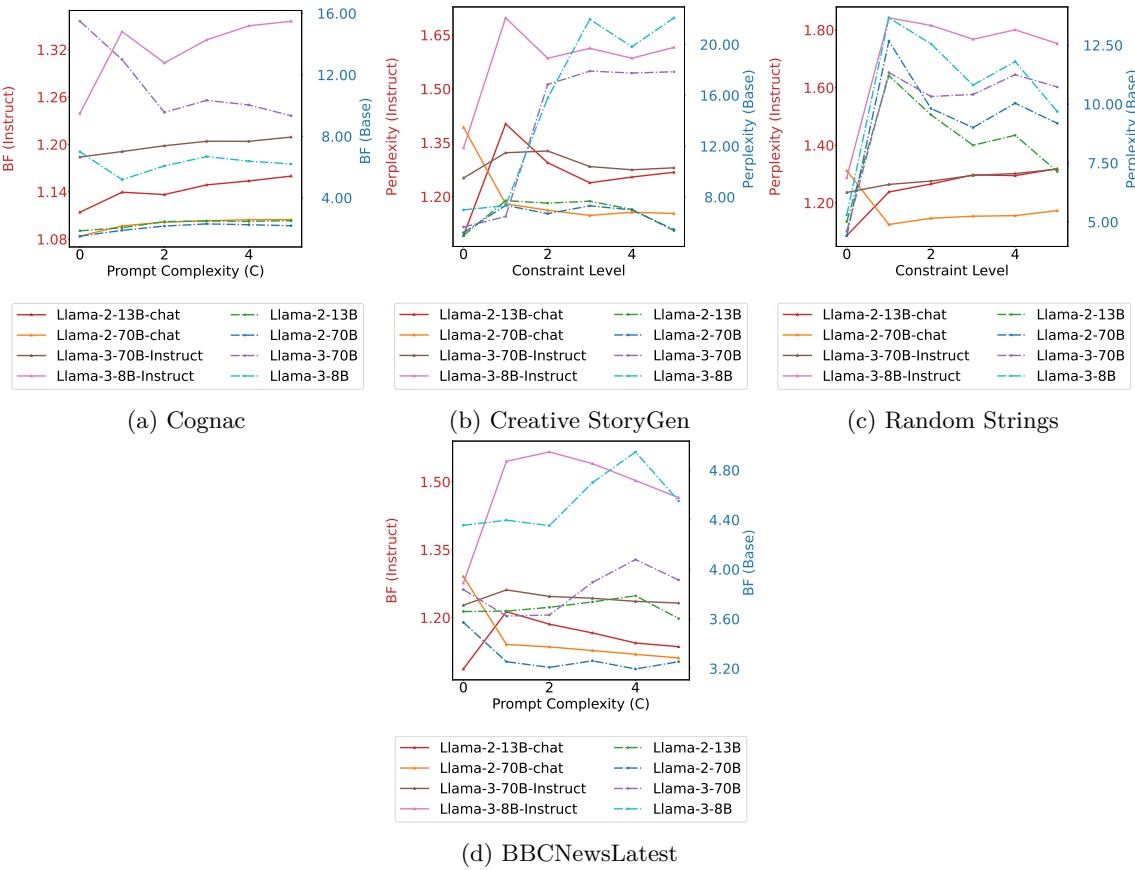

Figure 14: BF changes with prompt complexity ($C$) for Different Tasks. We can see prompt complexity affects BF in a task-varied way.

original paper. While a more direct proof using the weak law of large numbers is possible, we use Chebyshev's inequality to provide a more self-contained and accessible argument.

The key observation here is that under current computation architecture, the probability implemented by transformers are log-precision (Merrill & Sabharwal, 2023), and thus $|\log P(y_{1:N}|x;\theta)|$ is bounded (e.g., $|\log P(y_{1:N}|x;\theta)| \leq M$). For the truncated probability $\tilde{P}(y_{1:N}|x;\theta)$, we can essentially only consider the non-zero probability over the truncated vocabulary, and the same bound holds. Depending on the quantization scheme implemented, examples of $M$ include $32, 64$, etc.

We define the length-averaged *realized entropy* for a specific sequence $y_{1:N}$ as:

$$h_{\text{realized}}(y_{1:N}) \stackrel{\text{def}}{=} \frac{1}{N} \sum_{t=1}^{N} H(Y_t|[x, y_{<t}]; \theta) \tag{10}$$

where $H(Y_t|[x, y_{<t}]; \theta) = -\sum_{y \in V} \tilde{P}(y|[x, y_{<t}]; \theta) \log \tilde{P}(y|[x, y_{<t}]; \theta)$.

We aim to bound the probability that the NLL deviates from this realized entropy. Let $\Delta_N$ be the total difference between the log-probability and the realized entropy sum:

$$\Delta_N = \left(-\log \tilde{P}(y_{1:N}|x;\theta)\right) - \sum_{t=1}^{N} H(Y_t|[x, y_{<t}]; \theta) = \sum_{t=1}^{N} Z_t \tag{11}$$

where we define the single-step deviation variable $Z_t$ as:

$$Z_t \stackrel{\text{def}}{=} -\log \tilde{P}(y_t|[x, y_{<t}]; \theta) - H(Y_t|[x, y_{<t}]; \theta) \tag{12}$$

Note that $\Delta_N$ is a random variable formed by the sum of $Z_t$. To use Chebyshev's inequality, we need to calculate the variance of this sum:

$$\text{Var}(\Delta_N) = \text{Var}\left(\sum_{t=1}^{N} Z_t\right) = \sum_{t=1}^{N} \text{Var}(Z_t) + \sum_{i \neq j} \text{Cov}(Z_i, Z_j) \tag{13}$$

We now show that the covariance terms $\text{Cov}(Z_i, Z_j)$ are zero for all $i < j$. By definition, $\text{Cov}(Z_i, Z_j) = \mathbb{E}[Z_i Z_j] - \mathbb{E}[Z_i]\mathbb{E}[Z_j]$.

First, observe that the expected value of the deviation $Z_j$ at any step, conditioned on the prompt and generated history, is zero:

$$\begin{aligned}
\mathbb{E}[Z_j|[x, y_{<j}]; \theta] &= \mathbb{E}_{y_j}\left[-\log \tilde{P}(y_j|[x, y_{<j}]; \theta)\right] - H(Y_j|[x, y_{<j}]; \theta) \\
&= H(Y_j|[x, y_{<j}]; \theta) - H(Y_j|[x, y_{<j}]; \theta) = 0
\end{aligned} \tag{14}$$

This implies $\mathbb{E}[Z_j] = 0$ for all $j$. Thus, $\text{Cov}(Z_i, Z_j) = \mathbb{E}[Z_i Z_j]$.

For $i < j$, the value of $Z_i$ is fully determined by the history $[x, y_{<j}]$. We use the Law of Iterated Expectations, conditioning on the history up to step $j$:

$$\begin{aligned}
\mathbb{E}[Z_i Z_j] &= \mathbb{E}_{[x, y_{<j}]}\left[\mathbb{E}[Z_i Z_j|[x, y_{<j}]; \theta]\right] & \tag{15} \\
&= \mathbb{E}_{[x, y_{<j}]}\left[Z_i \cdot \mathbb{E}[Z_j|[x, y_{<j}]; \theta]\right] \quad (\text{since } Z_i \text{ is determined given } y_{<j}) & \tag{16} \\
&= \mathbb{E}_{[x, y_{<j}]}\left[Z_i \cdot 0\right] \quad (\text{by Eq. 14}) & \tag{17} \\
&= 0 & \tag{18}
\end{aligned}$$

Since all cross-terms are zero, the variance of the sum is simply the sum of the variances:

$$\text{Var}(\Delta_N) = \sum_{t=1}^{N} \text{Var}(Z_t) \tag{19}$$

Given the observation that transformer probabilities are computed with bounded log-precision (Merrill & Sabharwal, 2023), we have $|\log P(y|[x, y_{<t}]; \theta)| \leq M$ (For un-truncated $P$). Consequently, the random variable $Z_t$ is bounded, and its variance is bounded by a constant, denoted $C = (2M)^2$.

$$\text{Var}(\Delta_N) \leq N \cdot C \tag{20}$$

We can now apply Chebyshev's inequality to the length-averaged deviation:

$$P\left(\left|\frac{\Delta_N}{N}\right| \geq \epsilon\right) \leq \frac{\text{Var}(\Delta_N/N)}{\epsilon^2} = \frac{\frac{1}{N^2}\text{Var}(\Delta_N)}{\epsilon^2} \leq \frac{\frac{1}{N^2}(N \cdot C)}{\epsilon^2} = \frac{C}{N\epsilon^2} \tag{21}$$

Taking the limit as $N \to \infty$, the probability of deviation approaches 0. Thus, we have convergence in probability:

$$-\frac{1}{N}\log \tilde{P}(y_{1:N}|x; \theta) - h_{\text{realized}}(y_{1:N}) \xrightarrow{P} 0 \tag{22}$$

## I   Full Nudging Experiment Results

Due to space limits, we put the nudging experiment results for MMLU here. Though on MMLU, nudging does not reduce BF that quickly as over Just-Eval-Instruct, it does bring down BF of base models significantly, which verifies our hypothesis in § 6.

## J   BF and Information Density

Our BF measure can also be interpreted as capturing the information density that LLMs target to facilitate efficient communication (Genzel & Charniak, 2002; Jaeger & Levy, 2006; Levy, 2008; Mahowald et al., 2013; Meister et al., 2021; Verma et al., 2023). Prior work has leveraged both token-level log-probabilities and entropy rates ($\bar{H}$) as proxies for information density in human and machine communication. In Theorem 3.1, we formalize the connection between these views, showing that BF–defined as the exponentiated entropy rate–aligns naturally with this theoretical framework. Unlike prior studies focused primarily on linguistic theory or cognitive science, our work operationalizes this principle at scale across modern LLMs, linking information density to alignment training, decoding dynamics, and output variability in a unified analysis.

## K   Discussion: BF and Cross-Sample Diversity Measures

The diversity measures of Kirk et al. (2024) — Self-BLEU, expectation-adjusted distinct $n$-grams, and embedding cosine similarity — are statistics of a finite pool of $N$ generations sampled from the model. BF, in contrast, is a statistic of the underlying conditional distribution $P(Y_{1:N} \mid x; \theta)$ itself: it is the entropy rate of this distribution. Although our hybrid estimator (Equation (8)) computes BF from sampled sequences, the estimand is the distribution-level entropy rate, not a similarity score between sampled outputs.

The two quantities are correlated in practice but not interchangeable. Consider two stylized distributions over a single generation step. Distribution P places mass $\frac{1}{2}$ on each of two near-synonymous tokens whose continuations are nearly identical: BF = 2, but Self-BLEU between sampled pairs is high, indicating low cross-sample diversity. Distribution Q places mass 0.99 on one token and 0.01 on a token that triggers a wildly different continuation: BF $\approx$ 1, but the rare sample diverges sharply from the typical one, indicating high cross-sample diversity. The two measures order this pair of distributions in opposite directions. Empirically, our existing lexical-diversity analysis (§ M) confirms that BF and surface-level diversity correlate only weakly across the generation pools used in this paper, consistent with the formal argument.

We therefore view BF and cross-sample diversity as complementary rather than competing: cross-sample diversity asks "are these $N$ generations different from one another?"; BF asks "how concentrated is the underlying distribution from which any sample is drawn?". The position-wise, matched-length, and mid-generation analyses in the main text rely on the per-step distributional reading and do not have natural analogues in the sample-pool framework.

## L  Why Does BF Decrease? Disentangling Autoregressive Self-Narrowing from Alignment

A natural question – raised during review – is *why* BF often decreases over generation, and in particular why it also decreases for the RANDOM STRINGS task, where there is no semantic task structure that alignment, stylistic tokens, or distribution collapse could plausibly act on. We do not claim a theorem that BF must decrease at every token position; local increases can occur. Instead, we argue, and provide controlled evidence, that two effects have been conflated and should be separated:

- **Autoregressive self-narrowing (the trend).** Empirically, as an autoregressive model conditions on its own growing prefix, the next-token distribution often becomes more concentrated. This is a robust aggregate tendency in our experiments – for base and aligned models alike – and it does not require the context to be meaningful. It is consistent with autoregressive left-to-right training, information-processing intuitions, and our AEP analysis (Theorem 3.1), but the AEP result should not be read as proving monotone token-wise BF decrease.

- **Alignment (the level and steepness).** Alignment tuning lowers the absolute BF and often accelerates the early narrowing (consistent with our nudging experiments, § 6). It governs *how low* and *how fast*, rather than being the sole explanation for why decreasing BF is commonly observed.

Under this view the RANDOM STRINGS result is not contradictory but a useful demonstration of self-narrowing, precisely because no semantic structure is available to explain the decrease away.

**A controlled intervention: substituting external random tokens for the model's own prefix.** To separate the trend from alignment, we hold the model fixed and change only the *source* of the prefix it conditions on. In the baseline, the model generates normally and conditions on its own output. In the intervention, we replace the first $k$ tokens of context—which the baseline would have generated itself—with an equally long block of externally-sampled i.i.d. random tokens (content the model never produced), and then let it continue from token $k$ onward. We measure BF on a position axis aligned to the true generation index, so the baseline and the substituted runs are directly comparable. Figure 20 shows the result for three Base/Aligned pairs. Two observations are consistent across all models: (i) at the substitution point BF *surges* back into the high-BF regime, as external out-of-distribution content re-opens the consideration set; and (ii) BF then generally *decays again* under continued autoregression, retracing the same narrowing shape as the self-conditioned baseline. Because the only manipulated variable is the source of the prefix (self-generated vs. externally-supplied random tokens), this is an intervention rather than a correlation. It supports the view that the decreasing trend is a broad empirical regularity associated with autoregressive self-conditioning rather than an alignment artifact, while also showing that BF can be *increased* on demand by supplying unexpected content. Alignment changes the absolute scale (aligned/DPO models operate in a much lower BF band and re-collapse faster) but not the qualitative dynamics.

**A setting where information arrives over time: agentic environment feedback.** We also build a minimal one-turn agentic setting, inspired by controlled situational-understanding environments for testing state tracking in chat models (Yang & Ettinger, 2023), in which the model receives new external information mid-generation. Each prompt contains a task, a current environment state, and a partial plan; then we append one `Environment Feedback:` message and ask the model to revise the plan and choose the next action. The prompt template is:

```
You are an agent interacting with an environment.  Maintain a multi-step plan,
update it after each environment message, and choose the next action.

Task:  [task]

Current state:  <state>
```

```
Plan so far:  <plan>

Environment Feedback:  [feedback]

Given this feedback, revise the plan if needed and produce the next reasoning step
and action.
```

We instantiate four scenario families and cycle through them when generating prompts. The concrete setups are:

- **Chess endgame.** Task: play White in a simplified endgame with White king on e5, white queen on d4, and black king on g7. Plan: restrict the black king, then bring the white king closer before checkmate. Control feedback says the queen improved control of the seventh rank; adversarial feedback says the black king found an escape square and direct checks now risk stalemate or repetition.

- **Warehouse robot.** Task: move a fragile package from shelf A to packing station D. State: the robot is at shelf A, the package is secured, corridor B is open, and station D is available. Plan: move through corridor B and place the package on a padded tray. Control feedback says the robot reached corridor B and the package remains stable; adversarial feedback says a cart blocks corridor B and the grip sensor reports instability.

- **Python debugging.** Task: debug a small CSV-processing pipeline. State: the parser loads a CSV file, validates rows, and writes normalized records. Plan: reproduce the failing row, verify schema checks, then patch the narrowest failing component. Control feedback says the malformed timestamp is correctly rejected; adversarial feedback says the schema check passed, but the file can be empty and the parser silently returns `None`.

- **Search-and-rescue drone.** Task: navigate a drone to locate a missing person. State: the drone is in hallway H1, the target beacon is strongest toward room R3, and battery is at 62%. Plan: enter R3, scan, then return through H1 if the beacon weakens. Control feedback says the drone entered R3 and the path back remains clear; adversarial feedback says smoke filled R3, the beacon reflected from a metal door, and battery use increased.

Within each scenario, the task, state, and plan are identical across conditions; only the feedback field changes. The *control* condition reports normal progress consistent with the current plan. The *adversarial* condition introduces a new event that invalidates part of the plan. The *random-noise* condition fills the same feedback slot with random ASCII text of the same role in the prompt. We then compute BF on the model's next continuation using the same estimator as in § 4, and report the change in BF relative to the matched control. Figure 21 shows that adversarial and random-noise feedback *increase* BF relative to the matched control, with the effect largest for less-aligned models and compressed – occasionally near zero – for the most heavily aligned ones. This provides a concrete, repeatable answer to whether some inputs consistently raise BF: content that is *unexpected from the model's own predictive viewpoint* does.

**Connection to negation.** This mechanism also helps explain the prompt-complexity result reported in § F: in Cognac, increasing prompt complexity through *negation increases* BF. Negation is another instance of context that is hard to reconcile with the model's expectations. Across these three independent settings – substituted random tokens, adversarial environment feedback, and negation – we observe the same qualitative pattern: unexpected context can raise BF, while continued self-conditioning tends to lower it again.

**Scope of the causal claim.** We do not claim a mechanistic, circuit-level account of self-narrowing, nor a mathematical proof that BF must decrease monotonically. The intervention above is a behavioral one: by fixing the model and toggling only the prefix source, it separates the general autoregressive self-conditioning tendency from alignment more cleanly than purely observational evidence. Deeper mechanistic explanations – e.g., activation/norm dynamics or attention concentration – are a promising direction we leave to future work. Our claim is the more modest one that BF decrease is a robust aggregate tendency under autoregressive generation, while alignment governs its level and steepness.

## M   Discussion: Diversity and BF Correlation

Following the branching factor (BF) analysis in § 2, a higher BF suggests greater lexical diversity in finite samples. To examine the relationship between BF and traditional diversity metrics, we compute Distinct-N (Li et al., 2016), incorporating necessary LLM-specific adaptations (Tevet & Berant, 2021; Guo et al., 2024; Kirk et al., 2024). We then conduct a correlation analysis between Distinct-N and BF.

Our results, presented in Figure 22, show **no consistent correlation** between BF and Distinct-N. Depending on the model and task, the relationship can be strongly positive, strongly negative, or entirely absent (e.g., Llama-3-70B-Instruct on Cognac at Figure 22b). This empirical inconsistency highlights a fundamental conceptual point: *BF measures a property of the underlying probability distribution, whereas diversity metrics measure a surface property of finite samples.*

BF, as the exponentiated entropy, characterizes the "width" of the model's entire output distribution. In contrast, metrics like Distinct-N describe a small set of sampled outputs and are known to be unreliable proxies for distributional properties, being sensitive to confounding factors like generation length (Liu et al., 2022).[13] This distinction is critical, as two models can produce samples of similar diversity while having fundamentally different underlying distributions (e.g., with infinite KL-divergence), a nuance that BF captures but sample-based metrics miss. Therefore, our work focuses on probability concentration, measured by BF, as a more fundamental and insightful tool for understanding a model's generative process.

Viewing alignment through the lens of BF reduction provides a unified framework that explains several disparate observations: it clarifies how alignment shrinks the generative horizon, why aligned models are less sensitive to decoding methods, and how techniques like Chain-of-Thought stabilize generation by shifting information to low-BF regions. This focus on distributional properties aligns with emerging research highlighting the importance of a model's entropy in understanding and improving advanced reasoning capabilities Cui et al. (2025); Wu et al. (2025b).

## N   Confounder Investigation: Data Contamination

A potential confounder in our analysis is the influence of data contamination. If prompts closely resemble the training data (including pretraining and alignment tuning, i.e., "data contamination"), smaller BF values would be expected, and vice versa. To evaluate this, we use the Min-K% metric (Shi et al., 2024b), which quantifies the overlap between experimental prompts and training data. Following Shi et al. (2024b), we set $K = 20$ and compute the average log-likelihood for the minimum $K\%$ of tokens. Using these Min-K% values, we perform a linear regression with BF to assess their correlation. For each task-model pair, Signed $R^2$ values are reported to indicate the strength and sign (positive or negative) of the correlation.

The results of the Min-K% analysis are presented in Figure 23. Significant negative correlations between BF and Min-K% are observed for models such as Llama-3-8B-Instruct, Llama-3-70B-Instruct, and Llama-2-70B-Chat across several tasks. Conversely, Llama-3-8B and Llama-2-13B-Chat models exhibit positive correlations. For other models, correlations are notably weaker. Overall, there is no consistent correlation pattern between BF and Min-K% across datasets and models, suggesting that data contamination cannot fully explain our findings.

---

[13] While the EAD metric (Liu et al., 2022) mitigates this issue, it remains influenced by vocabulary size and is not model-agnostic.

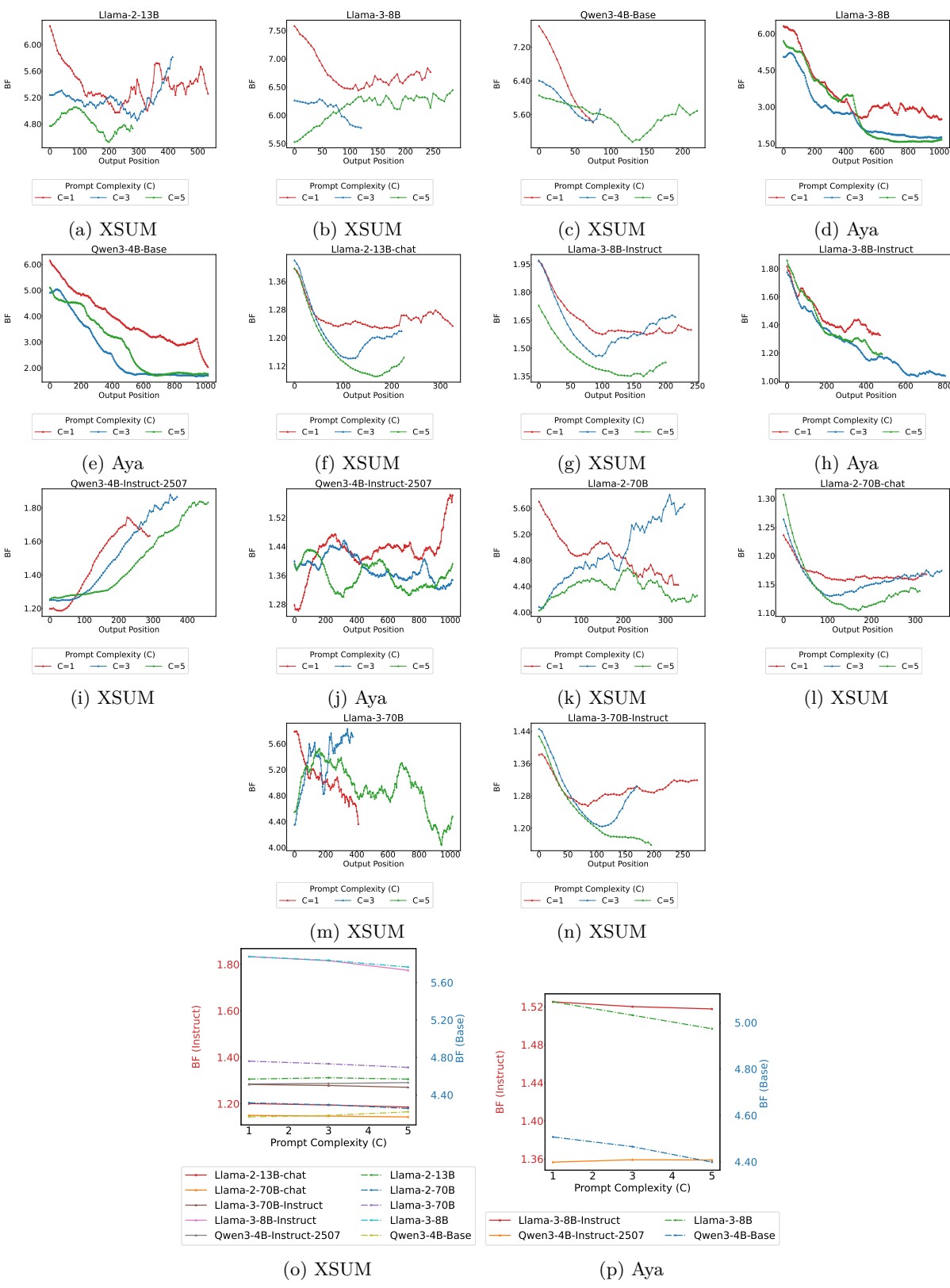

Figure 15: **Additional Verification on Summarization, Multilingual and Qwen Model family.** For better visualization, we compute the exponential moving averaged values of perplexity with the smoothing factor set as 0.1.

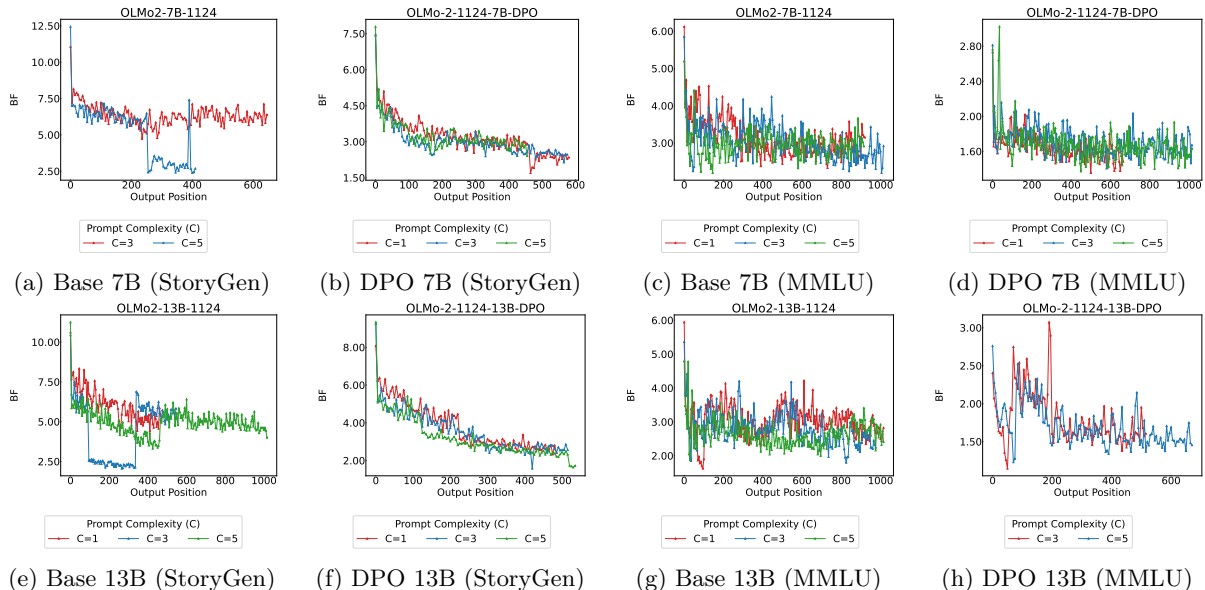

Figure 16: **BF Output Dynamic for OLMo-2 (7B & 13B) across alignment stages.** We compare Base and DPO models on Creative StoryGen and MMLU. Note that we treat DPO as the aligned model for OLMo-2, following the convention in (OLMo et al., 2024).

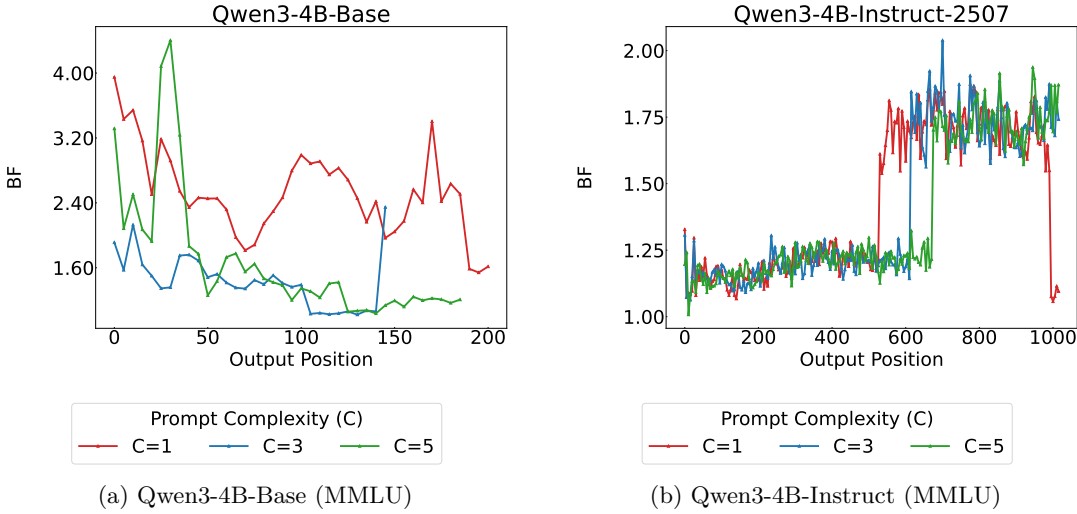

Figure 17: **BF Output Dynamic for Qwen3-4B on MMLU.**

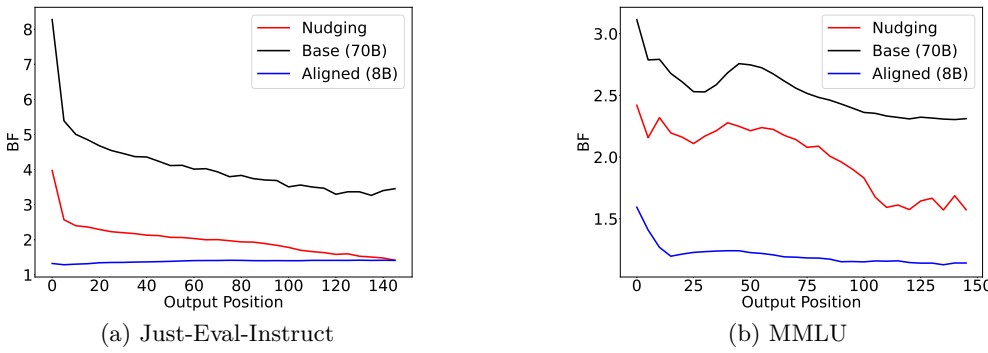

(a) Just-Eval-Instruct          (b) MMLU

Figure 18: Output Perplexity Dynamics in Nudging Experiments.

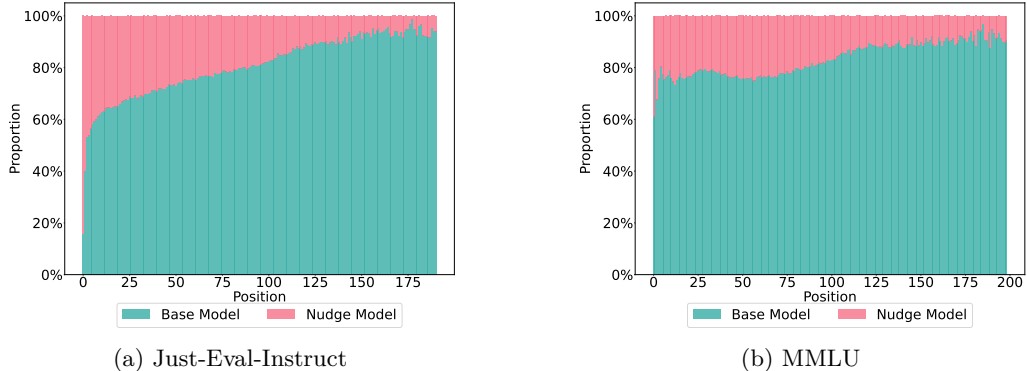

(a) Just-Eval-Instruct          (b) MMLU

Figure 19: Nudging Ratio Histogram.

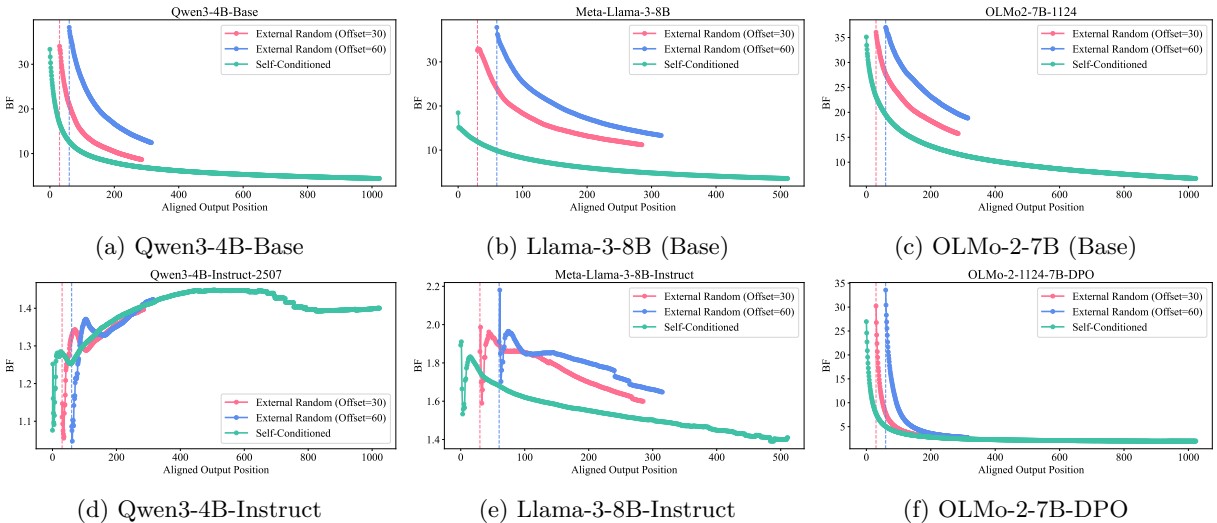

Figure 20: **Substituting external random tokens for the model's own prefix separates autoregressive self-narrowing from alignment.** The *Self-Conditioned* curve is ordinary generation, where the model conditions on its own output. The *External Random* curves replace the model's prefix with a block of i.i.d. random tokens up to a fixed cut point (30 or 60 model tokens, marked by dashed lines) and then continue; their *x*-axis is aligned to the true generation position. Supplying external content surges BF back up, after which continued autoregression generally drives it down again – in both Base (top) and Aligned (bottom) models. Alignment lowers the overall BF band and steepens re-collapse, but does not remove the aggregate decreasing tendency. We note one anomaly: Qwen3-4B-Instruct operates in an already very small BF range (roughly 1.1–1.4), where finite-sample estimation noise can noticeably affect the curve shape; although its tail still shows some decrease, we treat this panel cautiously and leave a sharper diagnosis to future work with access to intermediate checkpoints.

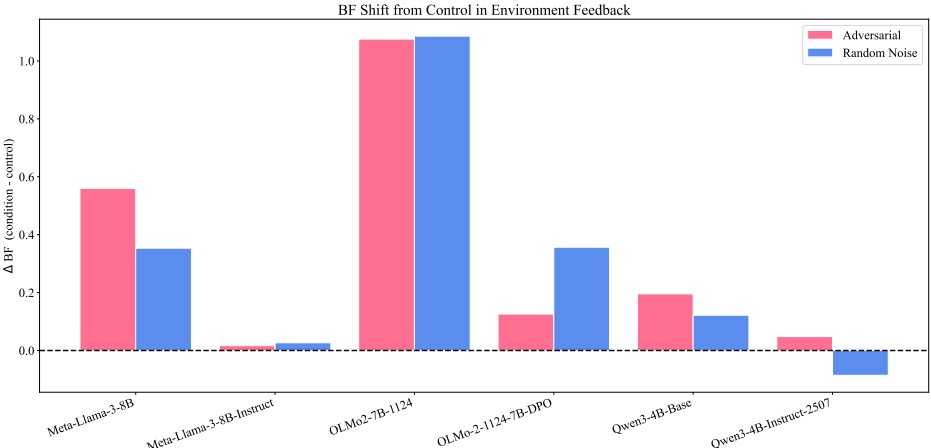

Figure 21: **Unexpected environment feedback raises BF.** Change in next-step BF relative to a matched "control" (progress-as-expected) feedback, for adversarial and random-noise feedback in a one-turn agentic setting. Unexpected content increases BF; the magnitude shrinks for more heavily aligned models, consistent with alignment compressing the available BF range.

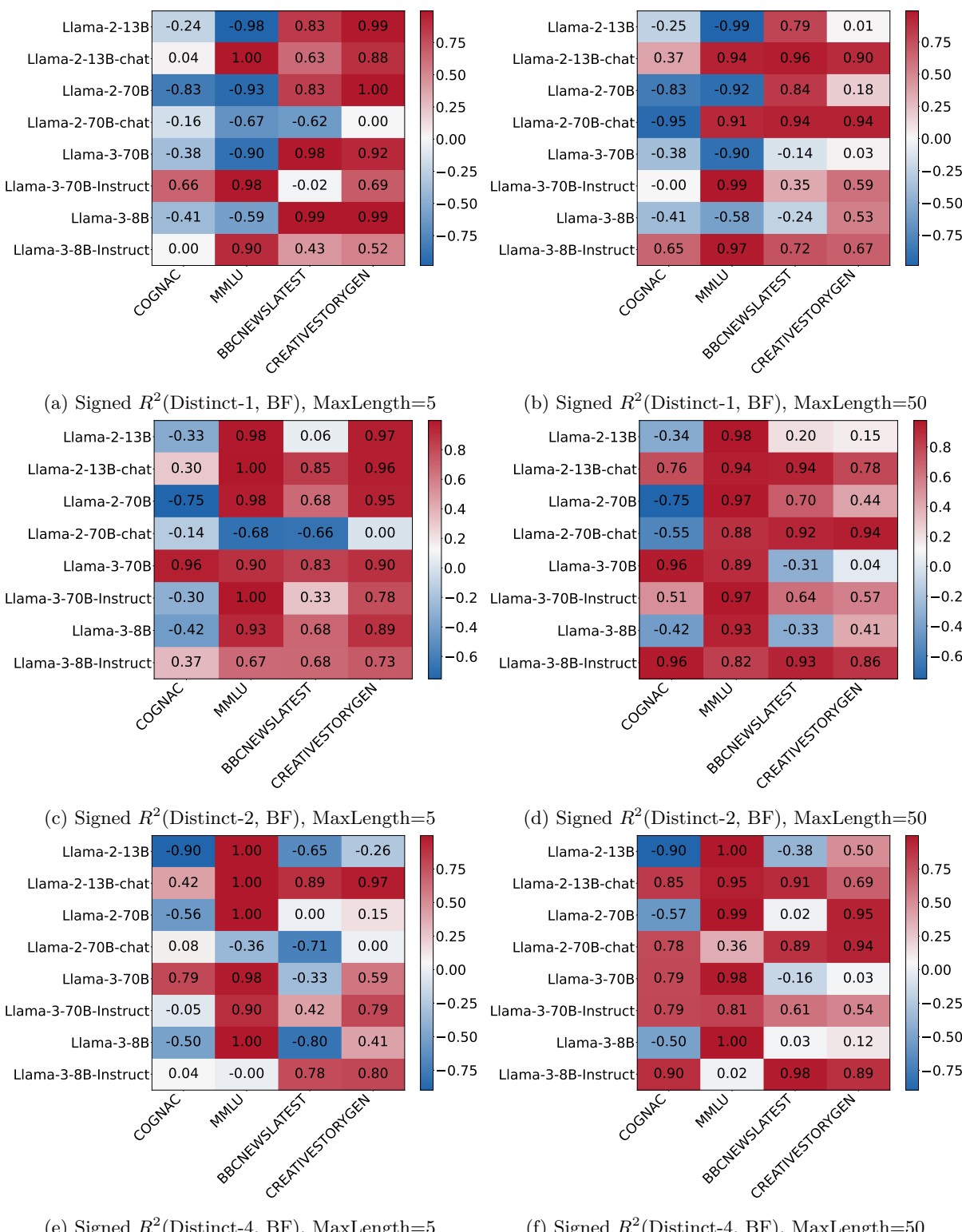

(a) Signed $R^2$(Distinct-1, BF), MaxLength=5

(b) Signed $R^2$(Distinct-1, BF), MaxLength=50

(c) Signed $R^2$(Distinct-2, BF), MaxLength=5

(d) Signed $R^2$(Distinct-2, BF), MaxLength=50

(e) Signed $R^2$(Distinct-4, BF), MaxLength=5

(f) Signed $R^2$(Distinct-4, BF), MaxLength=50

Figure 22: Correlational Analysis of BF and Distinct-N. We can find there is no consistent correlation between Distinct-N and BF.

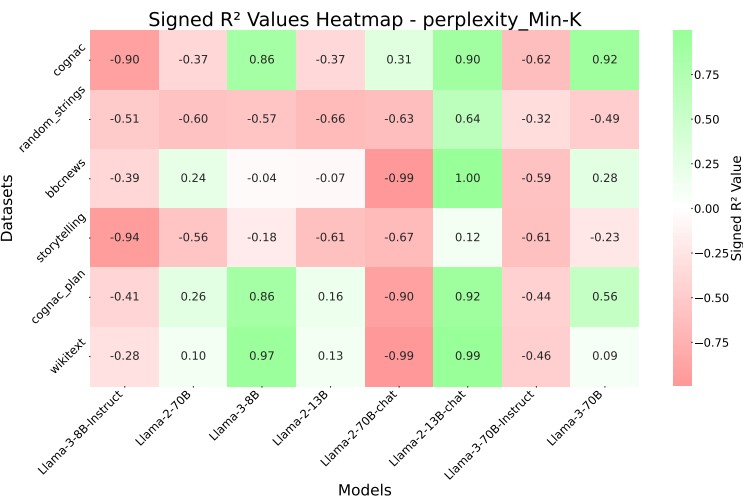

Figure 23: Signed $R^2$ values heatmap investigating correlation between BF and Min-K %.

