# OpenReview forum: "LLM Probability Concentration: How Alignment Shrinks the Generative Horizon"
_TMLR — Under review for TMLR_

### Review · Reviewer_V9az · 2026-03-27

**Summary Of Contributions:**

This paper systematically investigates the intrinsic mechanism behind the reduced diversity of aligned LLMs from a probabilistic perspective. It introduces the Branching Factor as a metric to quantify the effective number of branches during generation, by mapping sequence-level entropy to the width of an equivalent search tree, thereby characterizing the contraction of the model’s output space. The paper combines information-theoretic analysis with a scalable approximation based on using NLL as a proxy for entropy, enabling efficient estimation of BF for real-world models. Empirically, through comprehensive analysis across models, tasks, and alignment stages, they show that alignment significantly reduces BF by up to an order of magnitude, and that BF consistently decreases in the generation process. These findings provide a unified explanation for several phenomena, including the insensitivity of aligned models to decoding strategies, the increased stability of Chain-of-Thought reasoning, and the difficulty of late-stage exploration. Overall, the work offers an interpretable diagnostic framework that attributes diversity, stability, and alignment effects to probability concentration.

**Audience:**

Yes

**Audience Explanation:**

This work addresses a timely and broadly relevant question about the behavior of aligned LLMs and provides a unifying perspective that connects diversity, stability, and model abilities.

**Claims And Evidence:**

Yes

**Claims Explanation:**

The paper provides clear theoretical motivation and systematic empirical evidence across models, tasks, and decoding settings.

**Requested Changes:**

1. The paper could benefit from adding a dedicated limitations section to discuss potential shortcomings of the BF metric. For example, BF is an averaged quantity and may not adequately capture differences between distributions with distinct structural properties. In addition, the higher variance in early-stage sampling may introduce greater uncertainty into BF estimation. Furthermore, in certain deterministic tasks (e.g., mathematical computation), lower diversity in model outputs is expected and does not necessarily indicate a negative effect, which could nuance the interpretation of BF.

2. Since model generation typically exhibits higher variance in early stages and lower variance in later stages, the BF comparison for CoT versus non-CoT settings may not be entirely fair. It would strengthen the paper to include an additional experiment that computes BF under fixed-length truncation for both CoT and non-CoT generations. This could better support the claims and also help clarify whether the BF metric is truly length-invariant in practice.

3. The paper could be further strengthened by exploring the impact of other post-training strategies, such as PPO, to provide a more comprehensive understanding of how different alignment methods influence BF.

---

> ### Author Response · Authors · 2026-04-07
>
> Thank you for the careful and constructive suggestions. We have revised the paper to address all three requested changes (highlighted in blue color).
>
> 1. Limitations of BF. We have added a dedicated Limitations section (Section 9) and updated the Discussion section (Section 8). In particular, we incorporate the first two concerns you raised into the limitations: (i) BF is a first-moment summary and may not distinguish distributions with different corresponding tree structures, and (ii) estimation is less stable at short horizons, so we discuss the variance issue explicitly and motivate our hybrid estimator. Regarding your third point — that lower BF should not necessarily be interpreted negatively, since concentration can be desirable for tasks with a unique correct answer — we treat this as a conceptual clarification and address it in the Discussion section.
>
> 2. Fair CoT vs. non-CoT comparison. We added a matched-length control experiment in Section 4 using fixed-length truncation on MMLU. The new figure compares reasoning-oriented models and direct-answer instruct models at the same truncated output positions. We find that the reasoning-oriented models still exhibit consistently lower BF even at the earliest matched positions, which supports that the effect is not merely caused by longer generations and provides an empirical check that BF is robust to this length confound.
>
> 3. Other post-training strategies. We expanded Section 6 with an additional case study based on a publicly released RLHF pipeline from RLHFlow & OpenRLHF ([1, 2]), which includes SFT, DPO, PPO, and Iterative DPO checkpoints. This experiment broadens the stage-wise analysis beyond OLMo-2. We find that SFT still accounts for the largest BF reduction, while DPO and PPO have similar additional effects. Iterative DPO preserves more distributional breadth than the offline methods, which we discuss as a plausible consequence of its more online training loop.
>
> We appreciate these suggestions. They helped us make the paper more careful in its interpretation of BF and more complete in its empirical support.
>
> References:
>
> [1] Dong, Hanze, et al. "RLHF Workflow: From Reward Modeling to Online RLHF." TMLR 2024.
>
> [2] Hu, Jian, et al. "OpenRLHF: An Easy-to-use, Scalable and High-performance RLHF Framework." EMNLP 2025 Demo.

---

### Review · Reviewer_yrgK · 2026-05-03

**Summary Of Contributions:**

The paper proposes Branching Factor (BF), a length-normalized exponentiated entropy measure, as a unifying framework to quantify probability concentration in LLM generation. Through empirical analysis across various benchmarks and models, the authors report three findings: (1) alignment tuning reduces BF by a factor of 2 to 5 times on average, and by up to 10 times at early positions; (2) BF decays over the course of generation, and CoT models maintain lower BF even at matched lengths; and (3) base models can be steered into low-BF regimes via stylistic-token nudging. The paper provides a unified account of seemingly disparate observations, including reduced output diversity, decoding insensitivity, and CoT stability.


Strengths

-	Provides a comprehensive empirical sweep across multiple model families, scales, training stages (SFT, DPO, PPO, Iterative DPO), and tasks.

-	The matched-length CoT comparison and the stage-wise BF decomposition across post-training algorithms are novel analyses that disentangle effects prior work conflated.

-	The hybrid NLL/realized-entropy estimator enables tractable BF computation on long CoT generations.

Weaknesses

-	BF is mathematically equivalent to the length-normalized exponentiated sequence entropy, a standard information-theoretic quantity. The authors do explicitly disclaim novelty of the metric ("we do not propose BF as a novel mathematical metric”), but the abstract and introduction still frame BF as a contribution and the language of the motivation paragraph reads as if existing tools were inadequate. The disclaimer and the framing should be reconciled, preferably by leading with the equivalence rather than with a name-introduction.

-	The motivation for BF dismisses "token-level entropy" as too local without acknowledging that length-averaged entropy or entropy rate has been the standard aggregation and is what BF actually computes.

-	Two of the three headline findings substantially overlap with prior work that the paper cited in the paper. The paper does not articulate how its findings extend or differ from these prior work. Finding (1) overlaps with the central diversity-reduction result of [1] and is thematically related to [2] (which measured diversity reduction in a co-writing setting rather than directly on LLM output distributions). Finding (3) is the central hypothesis of [3], whose experimental design the authors directly reproduce.

-	BF inherits well-known limitations such as tokenizer dependence (which is non-trivial given that Figure 5a directly compares ratios across Llama-2/Llama-3/OLMo-2/Qwen3, all with different tokenizers), semantic-vs-surface insensitivity, and finite-sample tail underestimation. The paper does not discuss the first two limitations sufficiently; it documents the third in Appendix C but omits it as a caveat in the main text.

[1] Kirk et al. "Understanding the Effects of RLHF on LLM Generalisation and Diversity." ICLR 2024.

[2] Padmakumar & He. "Does Writing with Language Models Reduce Content Diversity?" ICLR 2024.

[3] Fei et al. "Nudging: Inference-time Alignment of LLMs via Guided Decoding." ACL 2025.

**Audience:**

No

**Audience Explanation:**

Most of the headline findings reproduce results already established in prior work that the paper cites. The reduction of diversity under alignment is the central result of [1], with related findings in a co-writing setting in [2]. The reduced sampling variance of aligned models is explicitly noted in [4] ("alignment can reduce sampling variance"), and the general insensitivity of LLM accuracy to decoding temperature in the 0–1 range is documented in [3] (though [3] does not explicitly contrast base vs. aligned models, which is a contribution this paper does add). The stylistic-token steering hypothesis with base-model uncertainty at those tokens is the central claim of [5], whose experimental design the authors explicitly reproduce. The matched-length CoT analysis and the stage-wise post-training decomposition are new, but they are individually narrow contributions and do not rise to the level of impact that would meaningfully inform the alignment, decoding, or reasoning communities of TMLR. Researchers familiar with this literature would likely view the paper as a re-measurement under a renamed metric rather than as offering new insight.

[1] Kirk et al. "Understanding the Effects of RLHF on LLM Generalisation and Diversity." ICLR 2024.

[2] Padmakumar & He. "Does Writing with Language Models Reduce Content Diversity?" ICLR 2024.

[3] Renze et al. "The Effect of Sampling Temperature on Problem Solving in Large Language Models." EMNLP 2024 Findings.

[4] Song et al. "The Good, The Bad, and The Greedy: Evaluation of LLMs Should Not Ignore Non-Determinism." NAACL 2025.

[5] Fei et al. "Nudging: Inference-time Alignment of LLMs via Guided Decoding." ACL 2025.

**Broader Impact Concerns:**

The paper includes a brief discussion of homogeneity bias from alignment tuning, which is appropriate. No additional concerns. The diagnostic nature of the work limits direct misuse risk, and the existing societal-impact paragraph adequately situates the findings.

**Claims And Evidence:**

Yes

**Claims Explanation:**

The empirical claims are supported by reasonably broad experiments across multiple model families and tasks, and the matched-length and stage-wise analyses are convincing. The mathematical derivations, including the BF definition, the chain-rule application, and the convergence proof of Theorem 3.1 are correct. The hybrid estimator design appropriately addresses the short-vs-long sequence tradeoff. However, the statement that BF provides a new lens unavailable from existing tools is not adequately supported, since BF is the exponentiated entropy rate, and most reported findings have been measured by prior work using equivalent or directly related entropy and diversity quantities. The contribution is better characterized as systematic re-measurement under one consistent metric plus a few new analyses, rather than novel discoveries.

**Requested Changes:**

-	Reframe BF as an interpretive repackaging of the length-normalized exponentiated sequence entropy, rather than as a novel metric. The disclaimer is good but is undercut by the abstract and introduction phrasing; revise these so that the equivalence to entropy rate is stated, cite the relevant information-theoretic background (entropy rate, perplexity), and revise claims that frame BF as capturing quantities previously unavailable from existing measures.

-	Substantially revise the differentiation from [1], [2], and [3], whose core findings overlap with this paper's headline results (and whose experimental setup, in the case of [3], is directly reproduced). Clarify which findings are genuinely novel versus which are re-measurements under BF of already-known phenomena.

-	Remove or rewrite the motivating critique that token-level entropy is too local, since the paper's own metric is the standard aggregation of the same quantity. Acknowledge length-averaged entropy or entropy rate as the established baseline.

-	Add an explicit discussion of BF's limitations including tokenizer dependence (especially for Figure 5a, where the heatmap directly compares BF ratios across model families with different tokenizers), insensitivity to semantic vs. surface diversity (BF cannot distinguish what semantic entropy in [4] was designed to capture), and the systematic underestimation from finite-sample MC entropy estimation that the paper itself documents in Appendix C. The last should be flagged as a caveat in the main text wherever cross-model BF comparisons appear.

[1] Kirk et al. "Understanding the Effects of RLHF on LLM Generalisation and Diversity." ICLR 2024.

[2] Padmakumar & He. "Does Writing with Language Models Reduce Content Diversity?" ICLR 2024.

[3] Fei et al. "Nudging: Inference-time Alignment of LLMs via Guided Decoding." ACL 2025.

[4] Kuhn et al. "Semantic Uncertainty: Linguistic Invariances for Uncertainty Estimation in Natural Language Generation." ICLR 2023.

---

> ### Author Response · Authors · 2026-05-17
> **Official Comment by Authors [Part.1]**
>
> We thank Reviewer yrgK for the careful read and the constructive list of requested changes. Below we address each requested change, and at the end we briefly discuss the audience question. All edits referenced below are now reflected in the revised PDF (highlighted in red color).
>
> **1. Reframe BF as an interpretive repackaging of the length-normalized exponentiated sequence entropy.**
>
> We agree, and we have revised the abstract and the introduction so that the equivalence to the entropy rate is stated up front and the framing of our contribution is unambiguously the *framework*, not the metric.
>
> - In the abstract, we now adopt the wording:
>     > "the lens of the Branching Factor (BF) -- the exponentiated length-averaged entropy of the model's output distribution, interpreted as the effective number of plausible next steps during generation"
>     >
>     rather than introducing BF as a token-invariant measure.
> - In the introduction, the motivation now focuses on:
>     > "we operationalize a principled distributional summary from information theory: the length-averaged entropy, or *entropy rate*, of the model's output distribution. We adopt the exponentiated entropy rate under the interpretive name Branching Factor (BF)."
>     >
>     We also retain the explicit disclaimer:
>     > "Crucially, we do not propose BF as a novel mathematical metric; rather, we introduce it as a unifying conceptual framework to quantify LLM probability concentration."
>     >
> This reconciles the disclaimer with the abstract/intro framing without removing BF as a useful interpretive name.

---

> ### Author Response · Authors · 2026-05-17
> **Official Comment by Authors [Part.2]**
>
> **2. Substantially revise the differentiation from relevant works.**
>
> Because our contribution is a *unifying framework*, overlap with these works at the level of individual phenomena is by design rather than a gap. In our revision (Related Works, Section 5.2, Section 7), we have clarified which findings from prior work are reproduced in our experiments and which aspects constitute extensions or novel contributions of our own.
>
> - *vs. Kirk et al. [1] and Padmakumar & He [2] (sample-level diversity reduction).*
>     In Section 5.2 we now write:
>     > "[the alignment-driven BF reduction] is consistent with the cross-sample diversity reduction independently documented by [1] and [2]; BF re-expresses the same phenomenon at the distributional, per-step level, which is what lets us connect it to decoding/inference-time behaviors (Section 5) and decompose it into stage-wise contributions (Section 6)."
>     >
>     A new appendix subsection ("BF and Cross-Sample Diversity Measures") makes the conceptual difference precise:
>     - Kirk-style measures (Self-BLEU, expectation-adjusted distinct-n, embedding cosine) are statistics of a finite pool of N samples; BF is a statistic of the underlying conditional distribution itself.
>     - The two can order distributions in *opposite* directions — we give a stylized example with two distributions P and Q where P has BF = 2 but high Self-BLEU (low cross-sample diversity), and Q has BF ≈ 1 but low Self-BLEU on rare samples (high cross-sample diversity).
>     - Empirically, our existing lexical-diversity correlation study (Appendix L) shows BF and surface-level diversity correlate only weakly across our generation pools.
>
>     Position-wise, matched-length, and mid-generation analyses in the main text all rely on the per-step distributional reading and have no natural analogue in the sample-pool framework.
> - *vs. Song et al. [5] and Renze [4] (decoding insensitivity / sampling variance).*
>     Prior work documents the *symptom* (aligned models are insensitive to decoding hyperparameters and have lower sampling variance). The Related Works section now states explicitly:
>     > "the BF framework pinpoints the underlying *mechanism*: the shrinking distributional landscape over the course of token-by-token generation. This conceptual lens allows us to further forecast and verify the same probability concentration effects in long-CoT models."
>     >
>     We have also rewritten Section 5.2 so that it leads with our prediction (long-CoT reasoning models, whose BF stays low throughout the trace, should show even smaller variance than direct-answer instruct models), with the [5] observation cited in support rather than in the lead position.
> - *vs. Fei et al. [3] (nudging).*
>     We adopt the same controlled experimental setup but use it to test a *different* hypothesis. Section 7 now reads:
>     > "[3] use their nudging setup to evaluate sample-level task performance for inference-time alignment, and our goal is *not* to re-explain that observation. Instead, we adopt the same controlled setup to *test our distributional hypothesis* about how probability concentration arises during alignment -- namely, that a small number of stylistic tokens is sufficient to steer a base model into the low-BF regime characteristic of aligned models. The BF measurement then provides the per-step distributional view (which prefix induces how much concentration, and where in the sequence) that sample-level metrics cannot resolve."
>
> We also note that the BF framework has begun to enable concrete downstream applications — inference-time methods for reasoning (DeepConf [9]) and open-ended generation (BACo [10]), and training-time methods for reasoning (EAD-RLVR [11]) and joint quality–diversity optimization (DARLING [12]) — which we cite at the end of the Related Works section as concurrent evidence that the framework is useful beyond the diagnostic findings reported here.
>
> **3. Remove or rewrite the motivating critique that token-level entropy is too local.**
>
> We completely agree with the reviewer's concern and the revised introduction no longer dismisses token-level entropy. We now operationalize length-averaged entropy / entropy rate as the principled distributional summary and present BF as its interpretive name. The remaining mentions of "model perplexity" and "n-gram diversity" are framed neutrally as measures that answer different questions (fit to a reference corpus; diversity of a finite sample set), not as inadequate alternatives.

---

> ### Author Response · Authors · 2026-05-17
> **Official Comment by Authors [Part.3]**
>
> **4. Explicit discussion of BF's limitations.**
>
> We have substantially expanded the Limitations section to cover the three concerns raised:
>
> - *Tokenizer dependence (and Figure 5a in particular).*
>     A new "Tokenizer Dependence and Cross-Family Comparison" subsection in Limitations states that raw BF magnitudes depend on the tokenizer and are not directly comparable across families, and that throughout the paper we either compare within a single family or report the within-family Base/Aligned BF *ratio*, which is tokenizer-invariant by construction. We have also updated the caption of Figure 5a directly:
>     > "Since both base and aligned models in each column share the same tokenizer, the within-family tokenizer scaling cancels in the ratio; cross-family ratio magnitudes should be interpreted as patterns rather than precise comparisons."
>     >
>     This makes the cross-family caveat visible at the point of consumption.
> - *Surface vs. semantic concentration.*
>     A new "Surface vs. Semantic Concentration" subsection in Limitations acknowledges that BF is a token-level distributional summary and is by construction insensitive to whether two distinct surface forms encode the same meaning, a distinction that semantic-uncertainty methods (e.g., Kuhn et al. [6]) are designed to capture. We frame the two as complementary, and note that the per-step concentration captured by BF accumulates over the sequence and manifests in measurable reductions in sample-level lexical and semantic diversity for aligned models, as documented in [1], [7], and [8].
> - *Finite-sample MC underestimation.*
>     This caveat now appears in both the main text and the limitations section. The estimator section in Section 4 explicitly states that a naive MC entropy estimator (without the inner full-vocabulary sum) systematically underestimates entropy at finite sample budgets because it misses the long tail, and that our hybrid estimator avoids this issue by combining inner-step exact entropy for short sequences with the AEP-justified NLL proxy for long ones. The Limitations section ("Limitations of BF Hybrid Estimator") additionally acknowledges that the finite-sample MC of the realized entropy itself underestimates the true entropy because the rare, high-entropy tail is under-sampled at finite M, with a pointer to Appendix C (estimated BF rises monotonically as M grows from 4 to 64). We therefore explicitly state:
>     > "absolute BF magnitudes should be read as lower bounds. Cross-model and base-vs-aligned ratios are more robust to this bias than absolute magnitudes, because the under-sampling has comparable scale across models with comparable effective output spaces"
>     >
>     which justifies our use of within-family ratios as the primary cross-model comparison throughout the paper.
>
> **On the audience question.**
>
> We respectfully push back here, while taking your suggestions seriously in the revision.
>
> Three of the aligned-model behaviors we explain in a single unified picture — (i) the alignment-induced diversity reduction documented by [1, 2], (ii) the decoding insensitivity / lower sampling variance documented by [4, 5], and (iii) the stylistic-token steering studied by [3] — are indeed already observed *individually* in the literature. Our work does not claim them as new findings (and the revised abstract/introduction make this explicit). But the unification itself is, we believe, the contribution that TMLR's audience benefits from: it lets these otherwise separately reported phenomena be diagnosed, predicted, and intervened on with a single distributional quantity. The remaining contributions — the dynamic BF decline within a single generation and the resulting CoT stability mechanism with a matched-length control (Section 5), the stage-wise dissection of OLMo-2 SFT/DPO and the algorithm-wise comparison across SFT/DPO/PPO/Iterative DPO checkpoints (Section 6), and the resampling experiment characterizing late-stage forking risk (Section 5.4) — are, to our knowledge, not previously reported. The other reviewer found these contributions valuable on their own.
>
> Concretely, follow-up work has already begun to build on this framework spanning both reasoning and open-ended generation, and both training and inference time — [9], [10], [11], [12] — which we take as initial external evidence that the unification is useful to the community.
>
> We believe a venue like TMLR is well-positioned to value unifying, mechanism-level analyses of phenomena that the field has previously reported only in isolation, and we have tried to make this positioning more visible in the revision. We are very grateful for the close reading; the requested changes have clearly improved the paper.

---

> ### Author Response · Authors · 2026-05-17
> **Official Comment by Authors [Part.4 Reference]**
>
> References
>
> [1] Kirk et al. "Understanding the Effects of RLHF on LLM Generalisation and Diversity." ICLR 2024.
>
> [2] Padmakumar & He. "Does Writing with Language Models Reduce Content Diversity?" ICLR 2024.
>
> [3] Fei et al. "Nudging: Inference-time Alignment of LLMs via Guided Decoding." ACL 2025.
>
> [4] Renze. "The Effect of Sampling Temperature on Problem Solving in Large Language Models." EMNLP 2024 Findings.
>
> [5] Song et al. "The Good, The Bad, and The Greedy: Evaluation of LLMs Should Not Ignore Non-Determinism." NAACL 2025.
>
> [6] Kuhn et al. "Semantic Uncertainty: Linguistic Invariances for Uncertainty Estimation in Natural Language Generation." ICLR 2023.
>
> [7] West & Potts. "Base Models Beat Aligned Models at Randomness and Creativity." COLM 2025.
>
> [8] Lake et al. "From Distributional to Overton Pluralism: Investigating Large Language Model Alignment." NAACL 2025.
>
> [9] Fu et al. "Deep Think with Confidence." ICLR 2026.
>
> [10] Wang et al. "Optimizing Diversity and Quality through Base-Aligned Model Collaboration." ICML 2026.
>
> [11] Yang et al. "Let it Calm: Exploratory Annealed Decoding for Verifiable Reinforcement Learning." arXiv:2510.05251, 2025.
>
> [12] Li et al. "Jointly Reinforcing Diversity and Quality in Language Model Generations." arXiv:2509.02534, 2025.

---

### Review · Reviewer_MvHN · 2026-06-04

**Summary Of Contributions:**

This work proposes branching factor (BF) -- a measure of "possible variability in the next token prediction".
From that perspective, the work shows that as the generation length increases, the model becomes more deterministic.
It contains a plethora of experiments that demonstrate this phenomenon from different angles.
While most of these results are not surprising and in-line with our existing understanding of post-training behaviour, they nevertheless, serve as a useful empirical analysis.
The primary drawbacks of this work are: 1) the somewhat surprising and then under-explained results from the random strings experiment as well as 2) the a lack of demonstrable reason "why" we observe the reduction in the BF, beyond discussing it.

**Additional Comments:**

Random strings experiment is particularly interesting because the BF still goes down which is surprising given the rest of the analysis.

If we attribute the reduction in BF to post-training alignment, outputs becoming more predictable (distribution collapsing), special tokens (Sure,...) kicking in, then none of that should hold in the case of random strings.

We can draw two non-exclusive conclusions from that:
1. reduction in BF random strings results from a different mechanism than for naturalistic data,
2. the insights provided in this work are correlated with the reduction in BF but not causal.

Perhaps there is some activation vanishing at play that would be worth checking.

For all experiment setups in this work, the task complexity becomes lower as the generation progresses. I wonder if it is possible to devise an experimental setting that would require the opposite -- where BF would need to increase for a "successful" generation. It could provide more insight into my two points.

Another way of looking at that is the work finds many ways of collapsing the distribution but not the opposite. Are there tokens that (semi-)consistently increase BF?

**Audience:**

Yes

**Audience Explanation:**

Steering the output distribution of LLMs is a critical area of research with diverse considerations and applications -- both towards safer and more predictable models, as well as towards more exploratory and creative domains.

**Claims And Evidence:**

Yes

**Claims Explanation:**

The work provides interesting and relevant insight. Most of the results align with our current understanding of model behaviour. However, the compilation of the analysis in a single body of work is in itself useful.

The work does not deliver the final answer -- why we observe the reduction in BF, and why the distribution shrinks. While this might make sense in intuitive, naturalistic settings, it is unclear why it also applies to the random strings experiment.

**Requested Changes:**

This is a summary; refer to the comment for more information.
- additional analysis on random strings and more discussion on how they don't match the core insight
- optionally, an experiment where a model would receive more information as the length increases

---

> ### Author Response · Authors · 2026-06-20
> **Official Comment by Authors [Part.1]**
>
> We thank Reviewer MvHN for the thoughtful and unusually generative review. The two concerns you raise -- (1) the "surprising and under-explained" random-strings result, and (2) the lack of a demonstrable reason *why* BF reduces -- point at the same underlying question. We ran two new controlled experiments during the discussion period that help clarify the issue and remove the apparent contradiction. All new material is in the revised PDF (new Appendix L section, highlighted in **teal** color), and the key takeaways are below.
>
> ## 1. Main clarification: BF reduction appears to be a robust autoregressive tendency; alignment shrinks the horizon
>
> The crux of your concern is exactly right: if BF reduction were *caused by* alignment, special tokens, or distribution collapse, then it should not appear for random strings. Our clarification is that these are two separable effects, and our original presentation might have conflated them:
>
> - **Autoregressive self-narrowing (the *trend*).** Empirically, as an autoregressive model conditions on its own growing prefix, the next-token distribution often becomes more concentrated, even when the context is not semantically meaningful. This is not a token-wise theorem -- local increases can and do occur -- but it is a robust aggregate tendency in our experiments (Section 4.1 and Appendix D). The random-strings result is therefore not an anomaly; it suggests that semantic task structure is not required for BF to decline. This view is consistent with autoregressive left-to-right training, information-processing intuitions, and our AEP result, but we do not claim that these arguments prove monotone BF decrease at every position.
> - **Alignment (the *level* and *steepness*).** Alignment tuning sharply lowers the absolute BF (Section 4.2 and Section 6) and often accelerates the early narrowing (via stylistic/template tokens, as in our nudging experiments, Section 6). It modulates *how low and how fast*, rather than being the sole explanation for why decreasing BF is observed.
>
> So there is no contradiction: BF decrease over generation is best understood as a strong empirical regularity of autoregressive self-conditioning, supported by but not rigorously implied by our theoretical discussion; alignment further shrinks the generative horizon on top of that tendency. To clarify this distinction, we have revised the relevant discussion to explicitly separate these effects and avoid suggesting that alignment alone accounts for the observed decrease.
>
>
> ## 2. New experiment (controlled intervention): substituting external random tokens isolates the trend from alignment
>
> To move from correlation toward a controlled test, we hold the model fixed and change *only* the source of the prefix it conditions on. In the baseline, the model generates normally and conditions on its own output. In the intervention, we **replace the first *k* tokens of context with an equally long block of truly i.i.d. random tokens** -- external, out-of-distribution content the model never produced -- and then let the model continue from there. BF is measured on a position axis aligned to the true generation index, so the baseline and the substituted runs are directly comparable. Details are in the new Appendix L, referenced from the revised Section 4.1.
>
> The result (Figure 20) is consistent across **base and aligned** models (Qwen3-4B Base/Instruct, Llama-3-8B Base/Instruct, OLMo-2-7B Base/DPO):
>
> - At the substitution point, BF **surges back up** to the high-BF regime -- the external surprise re-opens the consideration set.
> - It then **decays again** under continued autoregression, tracing the same narrowing shape as the self-conditioned baseline.
>
> This is an intervention, not merely a correlation: the only thing we change is the source of the prefix (self-generated vs. externally-supplied random tokens). It shows (a) the decreasing trend re-appears even after we force in content with *no* task structure, supporting the view that this trend is a broad empirical regularity of autoregressive self-conditioning rather than an alignment artifact; and (b) BF *can* be increased on demand by supplying unexpected/OOD content. Alignment changes the absolute scale (e.g., DPO/Instruct models operate at a much lower BF band and re-collapse faster), but not the qualitative dynamics.

---

> ### Author Response · Authors · 2026-06-20
> **Official Comment by Authors [Part. 2]**
>
> ## 3. New experiment: a setting where new information arrives over time, and unexpected feedback increases BF
>
> You raised a particularly helpful question: is there a setting where the model *receives more information as length increases*, and whether some inputs *(semi-)consistently increase BF*? To investigate this, we constructed a minimal one-step agentic environment. A model is first given a task and an associated plan, after which a single **"Environment Feedback:"** message is appended in one of three matched forms: **control** (progress proceeds as expected), **adversarial** (an unexpected event invalidates part of the plan), or **random noise** (the same feedback slot is filled with unrelated text). We then measure the BF of the subsequent reasoning step relative to the matched control condition.
>
> Across models, both **adversarial** and **random-noise** feedback lead to higher BF than the corresponding control condition (Figure 21). The effect is most pronounced in less-aligned models and becomes progressively compressed—approaching zero in some cases—for heavily aligned models, consistent with our broader observation that alignment reduces the available BF range. These results provide a concrete and repeatable answer to the question of whether certain inputs can reliably increase BF: they can. In particular, inputs that are "unexpected" under the model's ongoing generative process tend to increase BF.
>
> ## 4. Other points raised in the review
>
> **"Are there tokens that (semi-)consistently increase BF?" / a setting where BF must increase.** Beyond the agentic experiment above, we note that a result already reported in the paper points in the same direction.: in Cognac, increasing prompt complexity via **negation** *increases* BF (Appendix E, "Curious Case of Prompt Complexity"). Negation is another case of content that is hard to reconcile with the model's expectations. Across three independent settings -- substituted random tokens, adversarial environment feedback, and negation -- we see one consistent pattern: **unexpected context can raise BF; continued self-conditioning tends to lower it again.**
>
> **Is the reduction causal or correlational? Could "activation vanishing" be at play?** We are careful not to over-claim. We do *not* claim a mechanistic, circuit-level account, nor a theorem that BF must decrease at every position. What the prefix-substitution experiment (see Rebuttal Section 2) adds over the original observational evidence is a *controlled intervention*: fixing the model and toggling only the prefix source separates the general self-conditioning tendency from alignment. We agree a deeper mechanistic story -- the activation-vanishing hypothesis you suggest, or norm-growth/attention-concentration accounts -- is valuable, and we now flag it explicitly as future work. Our more modest claim is that BF decrease is a robust aggregate tendency under autoregressive generation, while alignment governs its level and steepness.
>
> **Practical implication.** One immediate corollary, which we now discuss in the paper, is that BF collapse can be temporarily mitigated by introducing out-of-distribution or otherwise unexpected content. Such inputs can push the model back into a higher-BF regime, which may be desirable in applications that benefit from diversity or exploration. However, because autoregressive self-conditioning subsequently re-concentrates the distribution, this intervention provides only a *local* and temporary re-expansion rather than a permanent remedy. We believe this limitation is itself an important practical takeaway.

---

### Author Response · Authors · 2026-04-23

Dear Action Editor,

I hope you are doing well. I am writing to kindly check on the current status of the review process for our submission.

We understand that the review process can take time, and we truly appreciate the effort from you and the reviewers. Please let us know if there are any updates or if there is anything we can provide to help facilitate the process.

Thank you very much for your time and support.

Best regards,

TMLR Submission7735 Authors

---

### Author Response · Authors · 2026-06-03
**Status Update on Revised Submission and Reviewer Responses**

Dear Action Editor,

We would like to briefly summarize the current revision status. We have now completed two rounds of revisions in response to the reviewers’ comments. The revisions responding to Reviewer V9az are highlighted in blue, and the more recent revisions responding to Reviewer yrgK are highlighted in red.

For Reviewer V9az, we added a dedicated limitations section, introduced a matched-length CoT vs. non-CoT comparison, and expanded the post-training analysis to include SFT, DPO, PPO, and Iterative DPO checkpoints. For Reviewer yrgK, we substantially revised the framing of BF as an interpretive use of exponentiated entropy rate, clarified overlap and distinctions with prior work, rewrote the motivation around entropy-based measures, and expanded the limitations discussion on tokenizer dependence, semantic vs. surface diversity, and finite-sample underestimation.

We believe these revisions address the requested changes from both reviewers and have made the paper more careful in positioning, interpretation, and empirical support. We would be very grateful if the review process could continue when convenient, and we are happy to provide any further clarification if helpful.

Sincerely,

The authors

---

### Author Response · Authors · 2026-07-14
**Summary of Revisions Across All Three Reviews (and a Note on Timeline)**

We thank the three reviewers for their thorough and constructive feedback, which has substantially improved the paper. Below we summarize each reviewer's requests, our responses, and where the changes appear in the revised PDF. Revisions are color-coded by reviewer: **blue** (V9az), **red** (yrgK), and **teal** (MvHN).

All reviewers found the unifying, distributional view of Branching Factor (BF) useful; the requested changes centered on (i) framing BF as an interpretive use of the entropy rate, (ii) treating BF's limitations and relation to prior work more carefully, and (iii) explaining *why* BF decreases, including for random strings. The revision addresses each.

## Reviewer V9az (blue)

- **Limitations of BF.** New Limitations section (Section 9) plus expanded Discussion (Section 8): BF is a first-moment summary that may not separate distributions with different tree structure; estimation is less stable at short horizons; and low BF is not inherently negative (concentration is desirable for unique-answer tasks).
- **Fair CoT vs. non-CoT.** Added a matched-length control on MMLU (Section 4). Reasoning models keep lower BF even at the earliest matched positions, so the effect is not a length artifact.
- **Other post-training strategies.** Section 6 now includes a public RLHF pipeline (RLHFlow & OpenRLHF) spanning SFT, DPO, PPO, and Iterative DPO: SFT drives the largest reduction; DPO and PPO add similar effects; Iterative DPO retains more distributional breadth, consistent with its more online loop.

## Reviewer yrgK (red)

- **BF as entropy rate.** The abstract and introduction now lead with the equivalence to the exponentiated length-averaged entropy, frame the contribution as the *framework* (not a new metric), and keep the disclaimer.
- **Differentiation from prior work.** In Related Works and Sections 5.2/7, we state which findings we reproduce vs. extend: BF re-expresses alignment-driven diversity reduction (Kirk et al.; Padmakumar & He) at the per-step level; pinpoints the *mechanism* behind decoding insensitivity (Song et al.; Renze); and reuses the nudging setup (Fei et al.) to test a distributional hypothesis. A new appendix subsection contrasts BF with cross-sample diversity measures, including an example where BF and Self-BLEU order distributions oppositely.
- **Token-level entropy critique removed.** The revised introduction no longer dismisses token-level entropy; it operationalizes length-averaged entropy / entropy rate as the principled summary and treats perplexity and n-gram diversity neutrally as measures that answer different questions.
- **Limitations.** We added subsections on tokenizer dependence (with an updated Figure 5a caption and reliance on tokenizer-invariant within-family ratios), surface vs. semantic concentration (complementary to semantic-uncertainty methods), and finite-sample MC underestimation (absolute BF magnitudes should be read as lower bounds; ratios are more robust).

## Reviewer MvHN (teal)
New material in Appendix L (Figures 20–21), referenced from the main text.

- **Disentangling the two effects.** We now separate two effects our original presentation conflated: *autoregressive self-narrowing* (a robust aggregate tendency for the next-token distribution to concentrate as the model conditions on its own growing prefix, even on non-semantic context) and *alignment* (which lowers the absolute BF level and steepens the early decline). Under this view the random-strings result is not contradictory but a clean demonstration of self-narrowing, since no semantic structure is available to explain the decline away.
- **Controlled intervention.** Holding the model fixed, we change only the prefix *source*: replacing the first k tokens with externally-sampled i.i.d. random tokens surges BF back up, then it narrows again — across base and aligned models. Since only the source changes, this is an intervention, not a correlation.
- **Information arriving over time.** In a minimal one-step agentic setting, adversarial and random-noise "Environment Feedback" raise BF vs. a matched control, most for less-aligned models — a repeatable case where inputs increase BF.
- **Scope.** We connect this to negation increasing BF in Cognac (Appendix E), and we avoid claiming a mechanistic account or a monotone token-wise theorem, flagging activation/norm dynamics as future work.

## A note on timeline

We note that the four-week window for reviewers' formal decision recommendations (ending Jul 2) has now passed. We fully respect everyone's time, and only wish to kindly ask the Action Editor about the current review status and when a decision might be expected. We remain glad to answer any remaining questions.

Sincerely,

The authors